# Structural basis of adaptor-mediated protein degradation by the tail-specific PDZ-protease Prc

Ming-Yuan Su[1,7], Nilanjan Som[2], Chia-Yun Wu[1], Shih-Chieh Su[1], Yi-Ting Kuo[3], Lu-Chu Ke[1], Meng-Ru Ho[1], Shiou-Ru Tzeng[3], Ching-Hao Teng[4], Dominique Mengin-Lecreulx [5], Manjula Reddy[2] & Chung-I Chang [1,6]

Peptidoglycan (PG) is a highly cross-linked, protective mesh-like sacculus that surrounds the bacterial cytoplasmic membrane. Expansion of PG is tightly coupled to growth of a bacterial cell and requires hydrolases to cleave the cross-links for insertion of nascent PG material. In *Escherichia coli*, a proteolytic system comprising the periplasmic PDZ-protease Prc and the lipoprotein adaptor NlpI contributes to PG enlargement by regulating cellular levels of MepS, a cross-link-specific hydrolase. Here, we demonstrate how NlpI binds Prc to facilitate the degradation of its substrate MepS by structural and mutational analyses. An NlpI homodimer binds two molecules of Prc and forms three-sided MepS-docking cradles using its tetra-tricopeptide repeats. Prc forms a monomeric bowl-shaped structure with a lid-like PDZ domain connected by a substrate-sensing hinge that recognizes the bound C terminus of the substrate. In summary, our study reveals mechanistic details of protein degradation by the PDZ-protease Prc bound to its cognate adaptor protein.

---

[1] Institute of Biological Chemistry, Academia Sinica, Taipei 11529, Taiwan. [2] CSIR-Centre for Cellular and Molecular Biology, Hyderabad 500007, India. [3] Institute of Biochemistry and Molecular Biology, College of Medicine, National Taiwan University, Taipei 10051, Taiwan. [4] Institute of Molecular Medicine, National Cheng Kung University, Tainan City 70456, Taiwan. [5] Institute for Integrative Biology of the Cell (I2BC), CEA, CNRS, Université Paris-Sud, Université Paris-Saclay, 91198 Gif-sur-Yvette, France. [6] Institute of Biochemical Sciences, College of Life Science, National Taiwan University, Taipei 10617, Taiwan. [7] Present address: Department of Molecular and Cell Biology, California Institute for Quantitative Biosciences, University of California, Berkeley, CA 94720, USA. Correspondence and requests for materials should be addressed to C.-I.C. (email: chungi@gate.sinica.edu.tw)

The bacterial cell walls contain peptidoglycan (PG or murein), a mesh-like sacculus that encases the cytoplasmic membrane to confer mechanical strength and shape to the cell. PG is composed of overlapping glycan chains, made of alternating β-1,4-linked *N*-acetylglucosamine (GlcNAc) and *N*-acetylmuramic acid (MurNAc) residues, which are cross-linked by short stem peptides[1]. In the Gram-negative bacterium *Escherichia coli*, the tetrapeptide consists of L-Ala-γ-D-Glu-*meso*-DAP-D-Ala (DAP is diaminopimelic acid) linked to the lactyl group of MurNAc, and cross-linking occurs directly between the carboxyl group of D-Ala of a (donor) stem peptide and the amino group of DAP of another (acceptor) stem peptide via an amide bond[1].

To maintain the net-shaped structure and its functional roles in bacteria, the synthesis and breakdown of PG must be intimately coupled to cell growth and division processes. It has been demonstrated that three *E. coli* PG hydrolases specific for the D-Ala-*meso*-DAP cross-link, MepS, MepM, and MepH, are

**Fig. 1** NlpI forms a tetrameric complex with Prc and enhances Prc-mediated substrate degradation. **a** Sedimentation velocity profiles of sNlpI (4.7S), Prc (5.0S), and the sNlpI-Prc complex (9.4S). **b** SEC-MALS analysis of the sNlpI-Prc complex. UV curve is plotted with fitted molecular weights. **c** and **d** Degradation of C- terminal **c** and N-terminal **d** 6×His-tagged sMepS (sMepS-6×His-C and N-6×His-sMepS, respectively) by Prc alone or with sNlpI, as monitored by SDS-PAGE assays. sMepS-6×His-C was used in all subsequent assays due to its superior stability. **e** Thermal shift melting curves (left) and their negative first derivatives (right) of Prc alone and with wild-type NlpI or the binding deficient mutant NlpI-QM (see Fig. 5 below). Melting temperature ($T_m$) values are shown within parentheses. **f** Degradation of native and 10 mM TCEP-treated denatured lysozyme by Prc alone or with sNlpI. **g** Degradation of sMepS by the sNlpI-Prc complex inhibited by increasing concentrations of the C-terminal peptide of MepS: NH₂-RYNEARRVLSRS-COOH (m). **h** and **i** SDS-PAGE assays monitoring degradation of MepS **h** and denatured lysozyme **i** by Prc-ΔPDZ without the PDZ domain

## Table 1 Crystallographic data collection and refinement statistics

| Crystal | sNlpI-Prc-K477A |
|---|---|
| *Data collection* | |
| Wavelength (Å) | 1.20 |
| Space group | $P2_12_12_1$ |
| Cell dimensions | |
| $a, b, c$ (Å) | 120.59, 146.73, 148.43 |
| $\alpha, \beta, \gamma$ (°) | 90, 90, 90 |
| Resolution (Å) | 30.00–2.30 (2.38–2.30)[a] |
| $I/\sigma I$ | 21.05 (2.04) |
| $R_{merge}$ | 0.067 (0.682) |
| $R_{pim}$[b] | 0.035 (0.35) |
| $CC_{1/2}$[c] | 0.955 (0.817) |
| Completeness (%) | 99.9 (100) |
| Redundancy | 4.5 (4.6) |
| *Refinement* | |
| Resolution (Å) | 30.00–2.30 (2.358–2.230) |
| No. of reflections | 105,902 (6,962) |
| Reflections used for $R_{free}$ | 5,466 (339) |
| $R_{work}$ | 0.2179 (0.2564) |
| $R_{free}$ | 0.2641 (0.3139) |
| No. of atoms | |
| Protein | 14,758 |
| Water | 946 |
| B-factors (Å$^2$) | |
| Protein | 44.87 |
| Water | 36.42 |
| R.m.s. deviations | |
| Bond lengths (Å) | 0.026 |
| Bond angles (°) | 1.78 |
| Validation | |
| Clash score | 22.07 |
| Rotamer outliers (%) | 11 |
| Ramachandran plot | |
| Favored/allowed/disallowed (%) | 97/2.2/0.38 |

[a]Highest resolution shell is shown in parenthesis
[b]$R_{pim}$ is the precision-indicating merging R, which describes the accuracy of the averaged measurement[29]
[c]$CC^{1/2}$ is the correlation coefficient between two random half data sets[30]

collectively required for bacterial growth and viability; mutants devoid of the three enzymes fail to incorporate new PG, leading to cell lysis[2]. Of these, MepS is an outer membrane (OM) lipoprotein whose cellular levels are high at the exponential phase of growth, but drop precipitously during the stationary phase; mutant cells with uncontrolled MepS levels do not grow on media of low osmolarity and form long filaments[2, 3]. The levels of MepS are regulated via proteolysis by a newly identified protease complex consisting of Prc, a soluble periplasmic PDZ-protease (also known as tail-specific protease, Tsp), and NlpI, an OM lipoprotein with tetratricopeptide repeats (TPRs)[3, 4]. NlpI serves as an adaptor protein that binds Prc to target MepS for degradation; however, interestingly, the latter two do not interact with each other[3]. Notably, the NlpI-Prc system completely degrades MepS into small fragments[3], which is unlike the other well-known C-terminal processing PDZ proteases that cleave the C termini of specific precursor protein substrates[5, 6].

Prc belongs to the large family of C-terminal processing proteases that recognize the C terminus of protein substrates[7]. Many of these PDZ-containing proteases, such as CtpB and D1P, are mostly involved in cleaving the C termini of specific protein substrates[5, 6]. Interestingly, Prc is able to process specific proteins by removing the C termini of FtsI or penicillin-binding protein 3 and also degrade protein substrates with nonpolar C termini or SsrA tags[8–12]. However, the structure of Prc has not yet known.

Moreover, it is not clear how the PDZ domain of Prc is involved in complete substrate degradation and how NlpI is involved in bridging the protease Prc and its substrate MepS. Here, we present the crystal structure of Prc in complex with NlpI. Combining structural analysis with biophysical, biochemical, and genetic assays, our results reveal mechanistic insights into the proteolytic degradation of MepS by the NlpI-Prc system.

## Results

**NlpI binding enhances the stability and activity of Prc**. We expressed and purified soluble MepS (19.0 kDa) and NlpI (33.8 kDa) proteins without their lipoprotein signal peptides (henceforth termed sMepS and sNlpI, respectively) and full-length Prc (77.6 kDa) based on previous studies[2, 9, 10, 12, 13]. Analytical ultracentrifugation (AUC) analysis suggested that recombinant sNlpI forms a homodimer and Prc is monomeric (Fig. 1a); the experimentally determined Mw's were $64.3 \pm 2.7$ kDa for NlpI and $67.8 \pm 5.5$ kDa for Prc. However, the purified sNlpI-Prc complex predominantly formed a 2:2 tetramer (Fig. 1a, b), with a determined Mw of $239.8 \pm 24.6$ kDa based on AUC and $230.2 \pm 8.7$ kDa based on size exclusion chromatography with multi-angle static light scattering (SEC-MALS). A gel-based assay showed that sMepS degrades slowly by Prc after 2 h; Prc alone also appeared somewhat unstable showing partial degradation in vitro (Fig. 1c, d). Thermal shift assays revealed the melting temperature ($T_m$) of Prc alone and Prc-sNlpI to be 40.1 and 53.6 °C, respectively, suggesting that the temperature-dependent stability of Prc is increased by forming a complex with sNlpI (Fig. 1e). Sodium dodecyl sulfate-polyacrylamide gel electrophoresis (SDS-PAGE) assay similarly showed that in the presence of sNlpI the stability of Prc was enhanced; importantly, sMepS with either a native or a 6×His-tagged C terminus was degraded rapidly by sNlpI-bound Prc within 10 min (Fig. 1c, d). The specific sMepS degradation activity of Prc alone and with sNlpI in 1:1 molar ratio were determined to be $0.580 \pm 0.102$ and $27.445 \pm 6.800$ nmol min$^{-1}$ per mg, respectively (see Methods), indicating a 50-fold enhancement of the specific activity mediated by sNlpI. sNlpI was also able to enhance the degradation efficiency of Prc against denatured-reduced lysozyme (Fig. 1f). Native lysozyme contains four disulfide links including one tethering the C terminus; reduced lysozyme is therefore expected to have a flexible C-terminal tail. Therefore, these results confirm that Prc alone recognizes the C terminus of substrate with low specificity[10]. Unlike the PDZ-protease CtpB[6], degradation of protein substrates by Prc requires the PDZ domain. A synthetic 12-residue peptide corresponding to the C terminus of MepS, which presumably acts as the PDZ ligand, was sufficient to inhibit MepS degradation in a concentration-dependent manner (Fig. 1g). Accordingly, Prc-ΔPDZ, without the PDZ domain, is inactive against MepS or denatured lysozyme (Fig. 1h, i). Together, these results highlight the key role of NlpI in enhancing substrate degradation by Prc, whose PDZ domain is indispensible for the proteolytic activity.

**Overall structure of the NlpI-Prc complex**. We determined the crystal structure of NlpI in complex with catalytically inactive Prc-K477A (hereafter termed the sNlpI-Prc complex), at 2.30 Å resolution (see Methods and Table 1). The asymmetric unit (ASU) contains two NlpI molecules forming a butterfly-shaped homodimer and two circular Prc molecules binding to each side of the symmetric NlpI dimer, which are angled away from the OM (Fig. 2a, b). The 223-kDa sNlpI-Prc tetrameric complex adopts an elongated shape, with overall dimensions of 180 Å × 75 Å × 80 Å. In the complex, the PDZ domain is facing towards the periplasmic space (Fig. 2a, b). The NlpI dimer interface involves mainly the swapped N-terminal helix h0 and a 10-residue linker

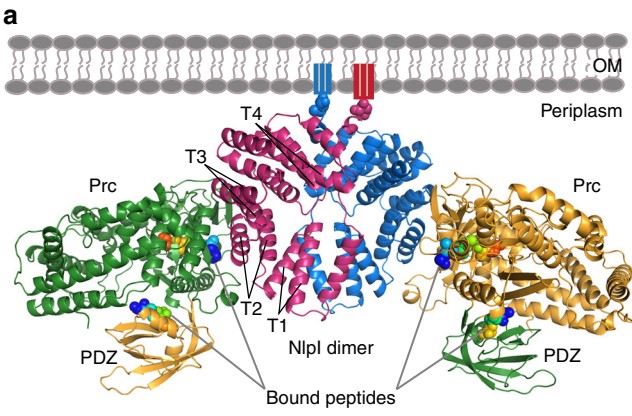

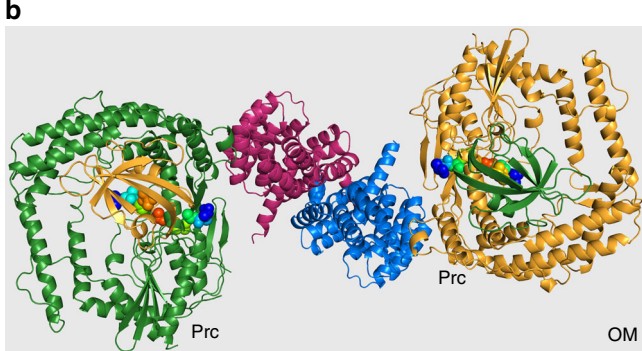

**Fig. 2** Overall structure of the sNlpI-Prc complex. **a** Ribbon diagram of dimeric NlpI bound to two Prc in different colors. The PDZ domain of Prc is highlighted in a contrasting color. The four tetratricopeptide repeats (TPR1–4) of NlpI are indicated (T1–4). The outer membrane (OM) and the lipid anchors are depicted in cartoons; the first residues linking to the lipobox cysteine, are shown in spheres. The four co-crystallized substrate peptides are rendered in rainbow-colored spheres. **b** View of the complex from the periplasmic space toward the OM (light gray background)

to helix h1 (Fig. 3a). The C-terminal helices are protruding from the panel surface in opposite directions; they are longer than those formed in the apo structure[13]. All four TPRs, each adopting a helix(a)–turn–helix(b) structure, are located on the outer edge of the dimer. NlpI binds to Prc mainly through the TPR2b helix, which contains four exposed acidic residues (D113, E117, D120, and E124) engaging in extensive electrostatic interactions with Prc (Fig. 3b and Supplementary Fig. 1). The combined presence of these residues is not found in other NlpI TPR motifs (Fig. 3c).

Prc forms a self-compartmentalized bowl-shaped body with a lid-like PDZ domain (Fig. 4); such monomeric structure is distinct from other C-terminal processing PDZ proteases[5, 6, 14]. The extended N-terminal and C-terminal helical domains (NHD and CHD, respectively), which associate through a pair of anti-parallel β-strands b1–b22 (Supplementary Fig. 2), forms a bowl-like structure with the protease module comprising a vault and the platform domain (Fig. 4a, b). The lid-like PDZ domain is attached to the bowl through a double hinge, consisting of two loops with conserved hydrophobic hinge residues L245 and L340 (Fig. 4c), which are connected to the vault helix h9 and the five-stranded β-sheet of the platform, respectively. In the structure, the PDZ domain is buttressed by the bowl by interacting with the NHD (see below). A conserved open substrate-binding passage containing the proteolytic site, as seen in the structure of active CtpB[6], is located to the side of the bowl near the hinged PDZ domain and is formed between the vault helix h9 and strands b2–b19–b20, and three parallel loops from the platform, which

harbor the catalytic K477-S452 dyad (Supplementary Fig. 2). A co-purified peptide is bound to the proteolytic site and co-crystallized serendipitously; the unidentified peptide is modeled as poly-Ala (Supplementary Fig. 3a, b). The NlpI-interaction domain (NID) contains helix h1, which packs against the vault strand b20, and helix h14 embedding in a large hairpin loop extending from the vault strands b19–20. Helix h1 makes electrostatic interactions with the TPR2b acidic residues of NlpI via basic residues R47, R51, and R484, while M493 and L494 of helix h14 make hydrophobic contacts with TPR1b–TPR2a of NlpI (Figs. 3b, 4b). An unidentified peptide is also bound to the peptide-binding site of the PDZ domain in each of the two Prc molecules (Supplementary Fig. 3c, d). Interestingly, the peptide bound to one of the PDZ domains could be tentatively modeled based on the electron-density map with the sequence LSRS-COOH, corresponding to the C-terminal four residues of MepS (Fig. 4c and Supplementary Fig. 3c).

**Compatible substrate and PDZ ligand-binding sites**. The proteolytic groove of Prc forms a small binding pocket for the substrate P1 residue, lined with hydrophobic residues V480 and F511 of the vault and A453, I456, and L400 of the platform (Supplementary Fig. 4a). By contrast, the binding site for the P2 residue is partially solvent exposed and formed solely by the vault residues F237, Y483, and Q505, which flank the entrance of the substrate-binding passage (Supplementary Fig. 4b). The binding sites for other prime and unprimed substrate residues are less defined and more solvent exposed. The peptide-binding groove of the PDZ domain of Prc also contains a small hydrophobic binding pocket, lined with residues I248, L252, V304, and I307, for the C-terminal residue and a solvent-exposed binding site composed of residues L245, V265, L340, and D342, for the penultimate residue of the PDZ ligand, which are compatible with the small hydrophobic P1- and the polar P2-binding pockets, respectively (Supplementary Fig. 4a, b). The matching substrate and PDZ ligand-binding pockets are also supported by using the peptide fragment LSRS-COOH, bound to one of the PDZ domain, as a model docked into substrate-binding pockets at the proteolytic site by superposition with the bound substrate peptide (Supplementary Fig. 4c). On the contrary, in CtpB the hydrophobic P1- and P2-binding pockets are notably smaller and bigger than those accommodating the C-terminal and the penultimate residues of the PDZ ligand, respectively (Supplementary Fig. 4d, e). The matching substrate/ ligand-binding pockets in Prc suggest that its PDZ domain may bind to the nascent C terminus formed after every cleavage of the substrate polypeptide chain, which may play a role in the substrate-degrading activity of Prc.

**TPR2 of NlpI is essential for interaction with Prc**. To determine the role of the four TPR2b acidic residues of NlpI, we performed pull-down assays using His-tagged sNlpI and the catalytically inactive mutant Prc-K477A with no tag. Removing two of the acidic residues was found to weaken the interaction between NlpI and Prc; removing three or all four of the acidic residues in the D113A/E117A/E124A triple mutant (sNlpI-TM) and the D113A/ E117A/D120/E124A quadruple mutant (sNlpI-QM) nearly abolished the binding (Fig. 5a). Accordingly, Prc incubated with sNlpI-QM did not show an increased $T_m$ shift and was inefficient in degrading sMepS (Figs. 1e, 5b). With sNlpI-QM, the specific degradation activity of Prc against MepS was determined to be $0.980 \pm 0.251$ nmol min$^{-1}$ per mg, comparable to that of Prc alone (see above). As soluble sNlpI and sMepS proteins used in SDS-PAGE analysis may not reflect their in vivo activity as OM-anchored lipoproteins, we examined the ability of plasmid-borne copies of NlpI's TPR2b mutants to functionally complement an

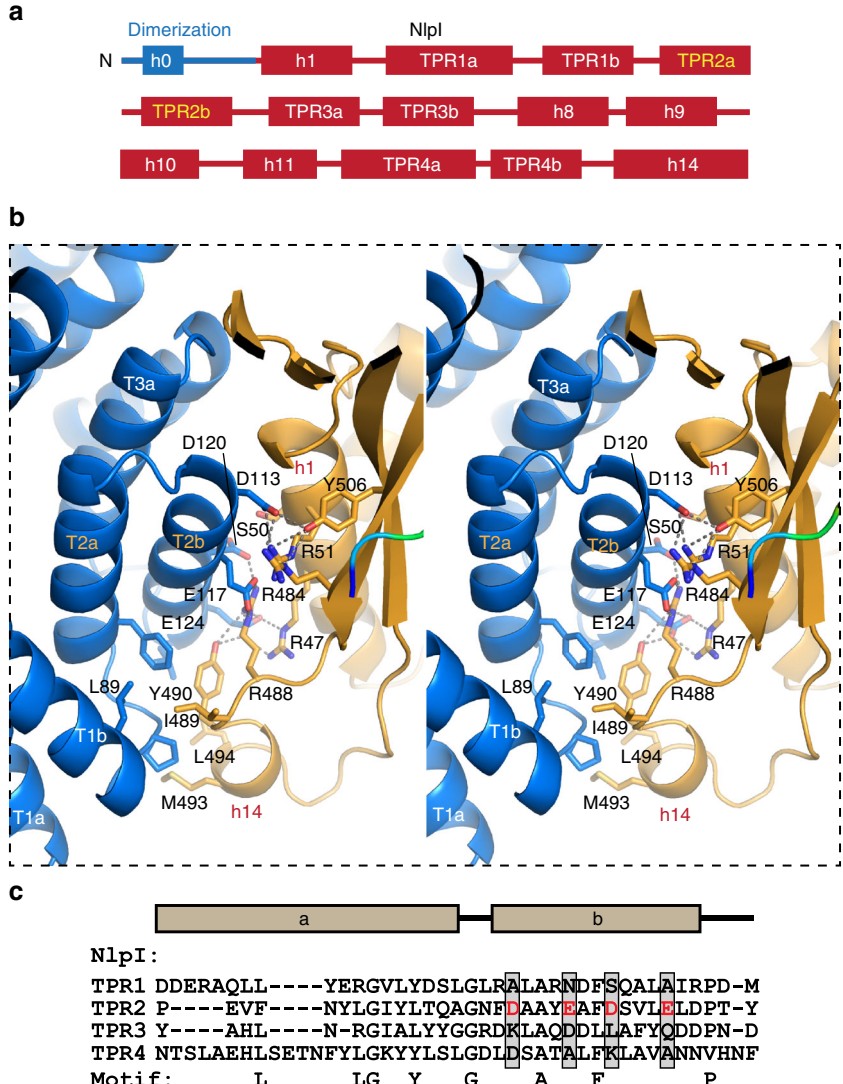

**Fig. 3** The binding interface between NlpI and Prc. **a** Domain organization of NlpI. The two TPR helices interacting with Prc are labeled in orange color. **b** Zoomed-in stereo view showing the interacting residues of NlpI (blue) and Prc (gold) in sticks. T2a/b and T3a indicate the TPR2a/b and TPR3a helices, respectively. **c** Sequence alignment of NlpI's TPRs highlighting the acidic residues in TPR2b

*nlpI* deletion strain. A deletion mutant of *nlpI* is known not to grow on media of low osmotic strength (LBON) (Fig. 5c). Leaky expression of wild-type (WT)-*nlpI*, but not *nlpI*-TM or *nlpI*-QM, was able to restore the NlpI function. Likewise, cell toxicity caused by overexpression of *nlpI*-TM or -QM was significantly reduced compared to WT-*nlpI*. The effect of the same NlpI mutations on MepS degradation in the *mepM* deletion strain deficient in an alternative endopeptidase was also examined; such strain would not be viable on rich media as NlpI overexpression results in depletion of MepS. Expression of *nlpI*-TM or -QM showed increasing viability in Δ*mepM* background, suggesting impaired MepS degradation in cells (Fig. 5c). Overall, these results support the critical role of the four acidic residues of TPR2b of NlpI in mediating Prc interaction.

**TPR1 of NlpI is involved in MepS binding**. The structure of the sNlpI-Prc complex permits the construction of a hypothetical docking model for interaction with MepS (PDB code 2K1G[15]) (see Methods). While detailed interactions cannot be verified without further experimental evidence and other models for ternary complex formation are possible, our MepS-docked model

is compatible with previous findings that MepS is a lipoprotein with an N-terminal lipid anchor, and that the substrate C terminus is recognized by Prc[9, 10, 12]. In this model, the docked MepS maintains its lipid anchoring topology while allowing its C terminus to access the substrate-binding tunnel of Prc to form a productive Michaelis complex (Fig. 6a, b). The resulting model reveals a prominent valley formed between the NlpI homodimer and Prc in the complex and flanked by NlpI's TPR1 and 2, the C-terminal helix h14, and Prc's β-sheet b15–16–19–20 (Figs. 4b, 6a), as the MepS-docking cradle with three open sides: the top side for the substrate's OM anchoring, the lateral side for substrate access to the valley, and bottom side for exposure of the MepS catalytic triad to the PG mesh in periplasm (Fig. 6a). Importantly, this model suggests that NlpI may bind to the N-terminal helix of MepS via the juxtaposed helices h1 and TPR1b formed by the two subunits of the NlpI homodimer (Fig. 6b).

To validate the docking model, we performed isothermal titration calorimetry (ITC) analysis of Prc titrated with NlpI, and MepS titrated with the Prc-NlpI complex or with NlpI alone (Fig. 6c–e). The analysis revealed that Prc binds to NlpI with very high affinity ($K_D < 10$ nM); the value could not be precisely

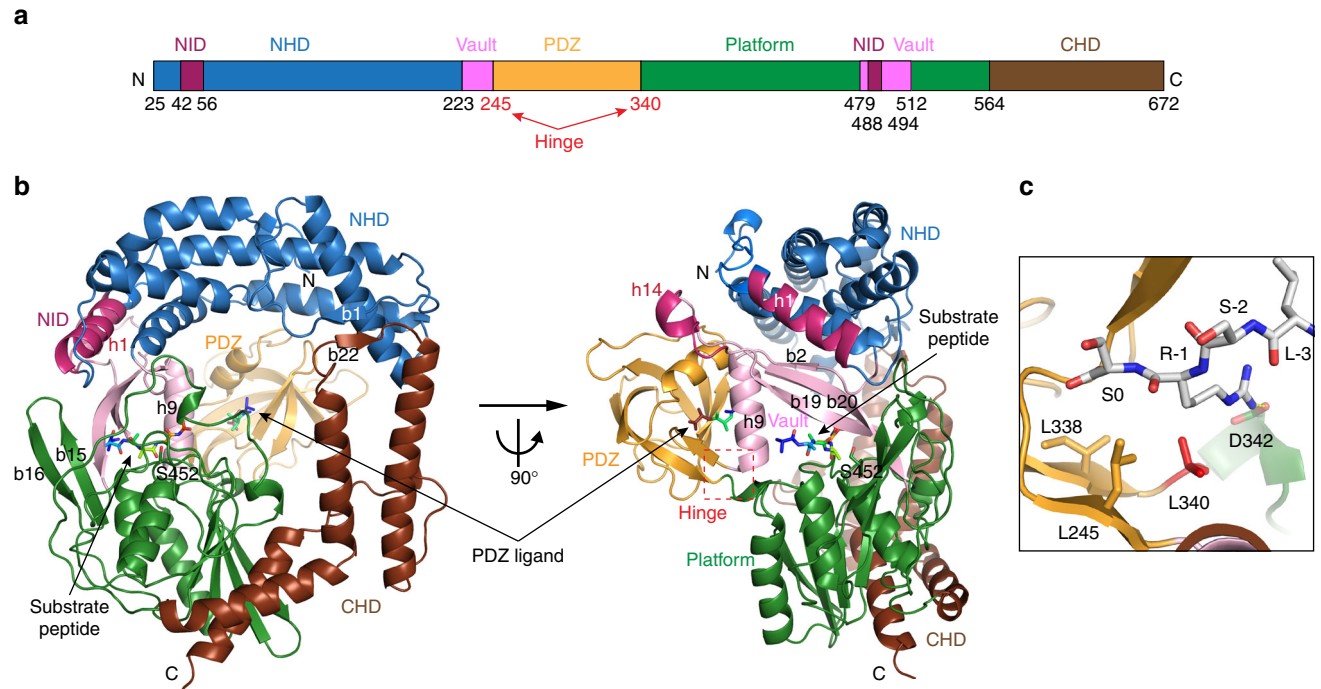

**Fig. 4** Overall structure of Prc. **a** Domain organization of Prc. NHD and CHD denote the Prc-specific N- and C-terminal helical domains, respectively. NID denotes NlpI-interaction domain. **b** Two orthogonal views of Prc. Only one of the two Prc molecules (chain C) in the structure is shown. **c** A zoomed-in view of the co-crystallized peptide bound to the PDZ domain and interacting with hydrophobic residues at the hinge. The peptide bound to chain C is modeled as the MepS C-terminal tetrapeptide LSRS-COOH, of which the penultimate residue (R-1) forms a hydrogen bond with D342

determined by ITC (Fig. 6c). The calculated $K_D$ values for sMepS binding to the sNlpI-Prc complex and to sNlpI alone were 2.09 and 0.75 μM, respectively (Fig. 6d, e). The comparable $K_D$ values suggest that the molecular surface of Prc in the NlpI-Prc complex may contribute little to the binding energy of MepS interaction. Indeed, ITC analysis of MepS titrated with Prc alone showed no apparent interaction between sMepS and Prc (Fig. 6f). Hence, focusing on characterizing the interaction of MepS with NlpI, we assessed the role of the helices h1 and TPR1b of the NlpI homodimer in mediating MepS interaction. Our ITC analysis showed that the monomeric mutant sNlpI-ΔN, devoid of residues M1-P36, does not bind to MepS (Fig. 7a, b). Interestingly, single mutations at residue L38 of helix h1 (L38A or L38C) or residue R82 of helix TPR1b (R82E) of NlpI, which are close to the N-terminal helix of the docked MepS, were sufficient to abolish MepS interaction without disrupting homodimer formation (Fig. 7c–f). By contrast, single mutation in TPR2b (A114W) only slightly weakened the binding (Fig. 7g). The MepS-binding effects of these NlpI mutations were also confirmed by in vivo assays, which showed that overexpression of both WT and NlpI-A114C decreased MepS level effectively to induce sickness in Δ*mepM* background; however, NlpI-L38C and NlpI-R82E failed to induce MepS depletion (Fig. 7h). All these mutants could nevertheless complement function of NlpI (Fig. 7i). Therefore, the ITC and in vivo results support the docking model and demonstrate the critical role of helices h1 and TPR1b in the NlpI homodimer in mediating MepS binding. These findings demonstrate that NlpI interacts with Prc by TPR2a/2b helices but with MepS by helices h1 and TPR1b, thus explaining the basis of NlpI binding to MepS in the absence of Prc[3].

**Hinge residues are involved in PDZ ligand sensing in Prc.** Structural comparison shows that in the structures of D1P and the resting form of CtpB, the PDZ domain forms close contact

with the strand b2 and helix h9, effectively blocking the substrate passage[5, 6]. By contrast, the PDZ domains in the two Prc structures in the ASU adopt a distinct swing-out open conformation (Fig. 8a). In the structure, the PDZ domain is stabilized by multiple salt-bridge interactions with helix h7 of the NHD while making contact with R232 of helix h9 at the top of the vault via another salt bridge (Fig. 8b, c). These observations suggest that the rigid-body rotation of the PDZ domain could be important for the activity of Prc. Previous study has identified the PDZ residue R168 of CtpB as a substrate sensor, which couples substrate-induced PDZ domain reorientation with protease activation by making contact with both the PDZ ligand and helix h9[6]. However, the corresponding residue K308 is partially disordered in Prc. Mutational analysis showed that Prc-K308W mutant marginally affected MepS degradation compared to the catalytic site mutant Prc-K477A (Fig. 9a, b). Prc-K308W could also complement Δ*prc* phenotype in vivo (Supplementary Fig. 5). Hence, K308 may not act as the substrate sensor in Prc.

In the Prc structure, two conserved residues L245 and L340 are located at the hinge region through which the protease body and the PDZ domain are connected. These two hinge residues make van der Waals contact not only with the penultimate residue of the bound PDZ peptide ligand, but also with helix h9 (Fig. 8c). Interestingly, the side chains of the corresponding CtpB hinge residues L199 and F111 adopt a solvent-exposed conformation in the apo resting state (Supplementary Fig. 6a). Hence, the hydrophobic hinge residues in Prc may play a role in sensing the PDZ ligand and contribute to stabilizing the ligand-bound PDZ domain in the open conformation. To determine their effect on MepS degradation activity, we have mutated residues L245 to alanine, but L340 to both alanine and glycine because L340 is located in a short loop (Figs. 4c and 8c). The single mutants Prc-L245A and Prc-L340A, especially the latter had moderately impaired MepS degradation activity (Fig. 9c). However,

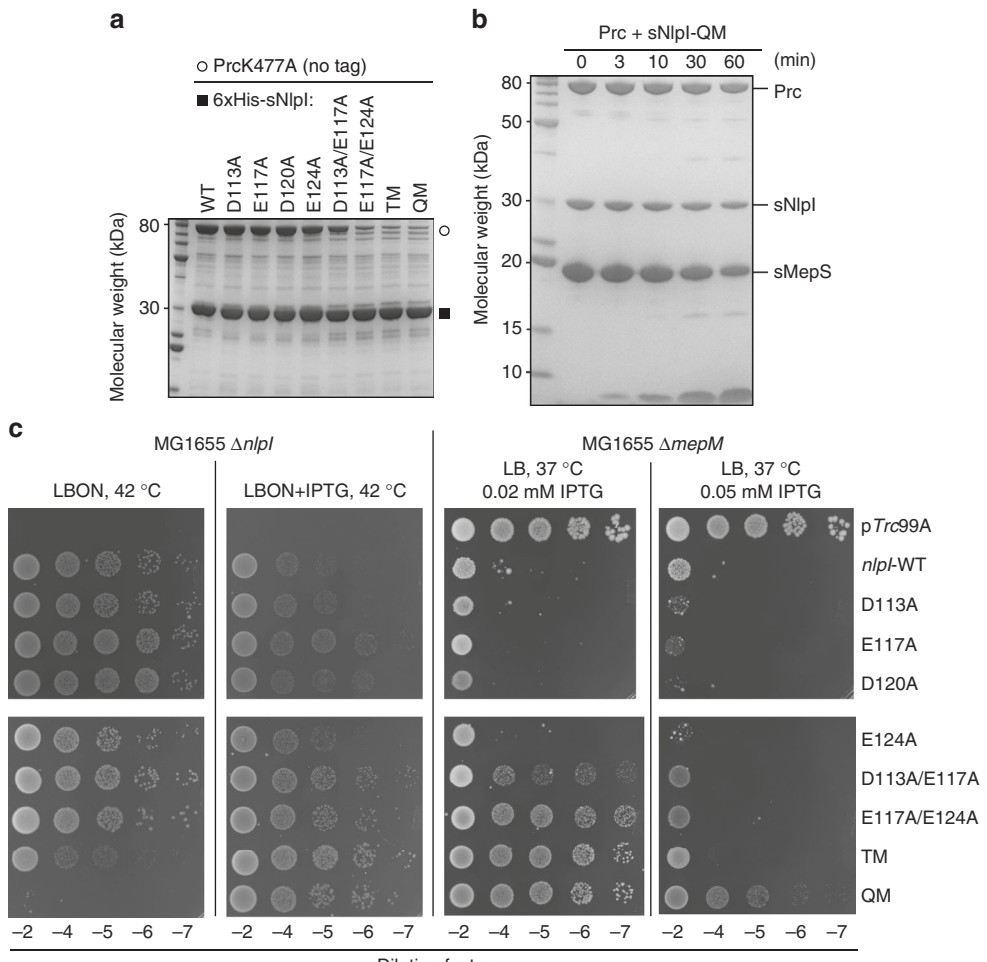

**Fig. 5** Mutational analysis of NlpI on Prc interaction. **a** Pull-down assays of Prc-K477A by wild-type (WT) or various mutants of 6×His-tagged sNlpI. **b** SDS-PAGE assay monitoring substrate degradation by Prc and sNlpI-QM. **c** Effects of NlpI mutations in cells of MG1655Δ*nlpI* or MG1655Δ*mepM* containing p*Trc*99A or pCHT93 carrying the WT or mutant NlpI derivatives. Cultures were grown in LB broth with ampicillin and 5 µl of various dilutions were spotted on indicated plates and grown overnight. TM, triple D113A/E117A/E124A mutant; QM, quadruple D113A/E117A/D120/E124A mutant

completely devoid of the side chain of the loop residue L340 at the hinge, Prc-L340G showed significantly abolished activity and Prc-L340G/L245A was almost completely inactive (Fig. 9d, e). There results support the critical role of the hinge region of Prc in PDZ ligand sensing, which is essential for degradation activity.

## Discussion
The structure of Prc revealed by this work is unlike that of D1P and CtpB; the latter proteases are known to cleave off the C termini of protein substrates. D1P forms a monomeric rod-shaped structure, whereas CtpB contains two dimerization motifs and forms a ring-like dimer[5, 6]. Here, it is revealed that mono-meric Prc forms a self-compartmentalized bowl-like structure with a hinged lid-like PDZ domain; the unique NHD and CHD of Prc constitute most of the bowl rim (Fig. 4). Moreover, unlike CtpB, the binding pockets in the proteolytic and PDZ ligand-binding sites of Prc are compatible in shape and polarity (Sup-plementary Fig. 4). Importantly, the ligand-bound PDZ domain is shown in the structure leaning against the NHD of Prc, yielding large outward rotational movement (Supplementary Fig. 6b). By contrast, both D1P and CtpB lack the extended structure of NHD; the PDZ domains of D1P and CtpB in the active or resting structures show that their PDZ domains only maintain contact

with the helix h9, allowing limited en bloc movement (Supple-mentary Fig. 6b). A similarly large rotation of the ligand-bound PDZ domain in D1P or CtpB would be energetically unfavorable in solution because all non-covalent interaction between the PDZ domain and the protease body would be lost. It should be noted that the functional role of the PDZ domains in Prc and CtpB are different. In CtpB, deletion of the PDZ domain yields a con-stitutively active C-terminal processing protease with no substrate selectivity[6]. However, the PDZ domain of Prc is absolutely required for substrate-degrading activity (Fig. 1). Based on the present results, it may be hypothesized that the large PDZ movement induced by ligand sensing and facilitated by the unique bowl-like structure of Prc may provide mechanical pulling force essential for degradation activity, enabling the protease to completely degrade protein substrates rather than cutting the C termini of substrate precursor proteins as CtpB or D1P.

The substrate adaptor NlpI is an OM-anchored protein with four TPRs, each repeat unit forming a helix–turn–helix motif. TPR motifs are known to mediate protein-protein interaction[16]. NlpI assembles into a symmetrical homodimer with all the TPRs exposed[13]. Here we show that NlpI uses non-overlapping sets of TPR and non-TPR helices to mediate helix–helix interaction with the effector Prc and the substrate MepS. Notably, the TPR2 unit mediating Prc interaction is uniquely rich in acidic residues.

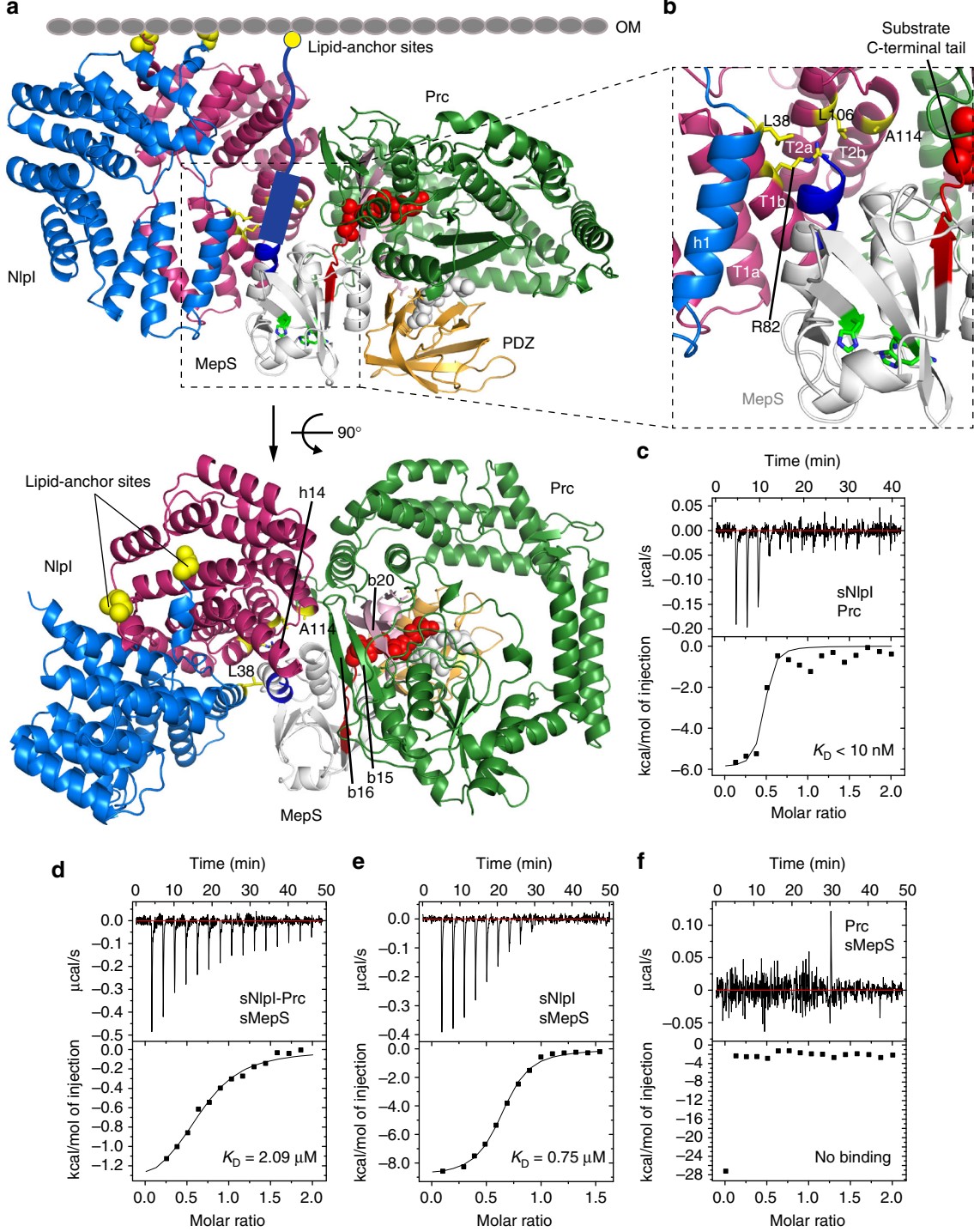

**Fig. 6** Analysis of MepS docking to the NlpI-Prc complex. **a** Two orthogonal views of MepS (PDB code 2K1G) docked to the structure of the sNlpI-Prc complex. Left: view parallel to the outer membrane (OM; depicted by cartoon). The N-terminal residues linking to the lipobox Cys are shown in yellow spheres. The predicted N-terminal coil and helix of MepS, missing in the NMR structure, are depicted by cartoons. The C-terminal coil of docked MepS is colored in red to highlight its close proximity to the substrate entrance pore of Prc, where a bound peptide is shown in red spheres. The catalytic Ser–His–His triad of MepS is highlighted in green. Right: view of the model from the OM toward the periplasmic space. Only one complexed Prc (chain C) is shown. **b** Zoomed-in view showing the putative MepS binding site of NlpI. **c** ITC analysis characterizing the interaction of Prc with sNlpI. **d–f** ITC analysis of sMepS with the sNlpI-Prc complex **d**, sNlpI alone **e**, and Prc alone **f**. Raw data (top) and binding isotherm derived from the integrated heat (bottom) are shown. The $K_D$ values are indicated

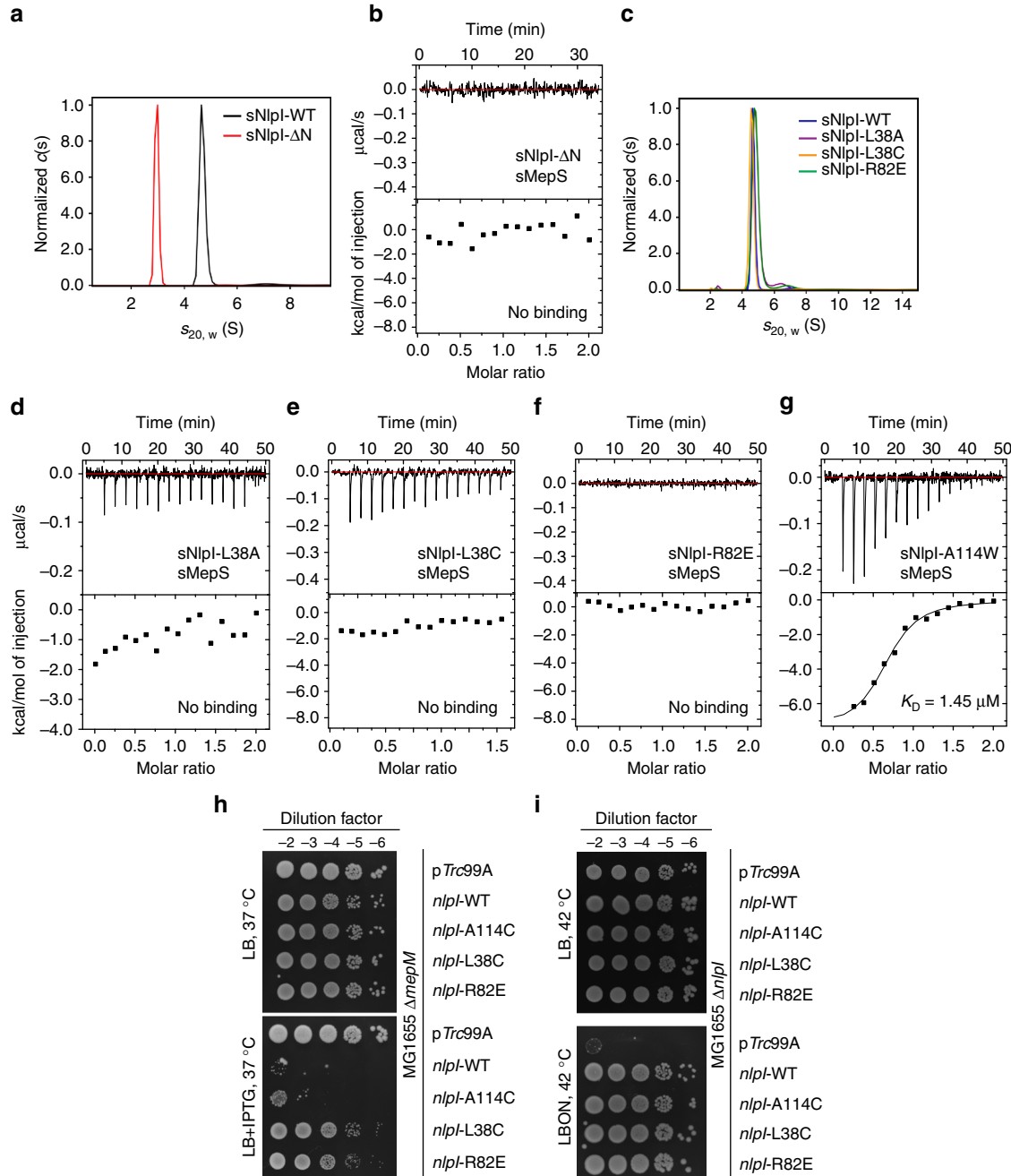

**Fig. 7** Mutational analysis of NlpI-MepS interaction. **a** Sedimentation velocity profiles of dimeric wild-type sNlpI and monomeric sNlpI-ΔN. **b** ITC analysis of the interaction of sNlpI-ΔN with sMepS showing no binding. **c** Sedimentation velocity profiles of sNlpI mutants showing dimer formation. **d**–**g** ITC analysis of the interaction of sMepS with sNlpI-L38A **d**, sNlpI-L38C **e**, sNlpI-R82E **f**, and sNlpI-A114W **g**. **h** and **i** Effect of *nlpI*-L38C overexpression in cells of MG1655Δ*mepM* **h** or MG1655Δ*nlpI* **i** containing p*Trc*99A or pCHT93 carrying the WT or mutant *nlpI* derivatives. Viability assays were done similarly as in Fig. 5c, with or without 0.05 mM IPTG

Therefore, our results indicate that the NlpI homodimer acts as hub to coordinate the assembly of multi-protein complexes via specific sets of multiple TPR motifs. However, it is currently unknown whether NlpI also participates in the assembly of other protein complexes.

In the structure, helix TPR2b of NlpI packs against the three-stranded vault sheet covering the substrate passage together with the NHD helix h1. Interestingly, the corresponding three-stranded sheet in CtpB is packed against an intermolecular four-helix bundle formed upon homo-dimerization[6]. By associating with Prc, the NlpI homodimer contributes an array of

helices, effectively extending the β-sheet surface of the vault of Prc to form a substrate-docking cradle. Therefore, it is conceivable that binding of NlpI likely increases the substrate specificity of Prc.

The overall structure of the sNlpI-Prc complex shows interesting features reminiscent of some specialized lever devices such as a wall-mount bottle opener and a yarn organizer. For example, the complex structure contains a fulcrum-like substrate-binding passage open on two sides: the side open from the exterior forms a cradle for the substrate load; the other side leading to the interior center of the bowl-shaped Prc is attached to a hinged

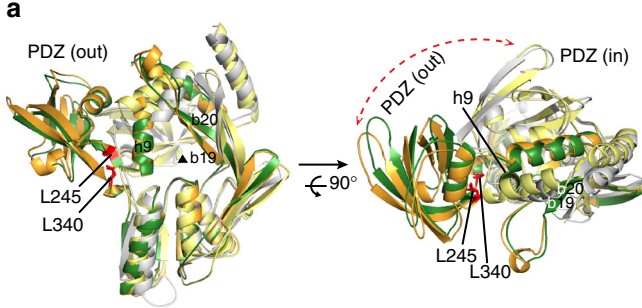

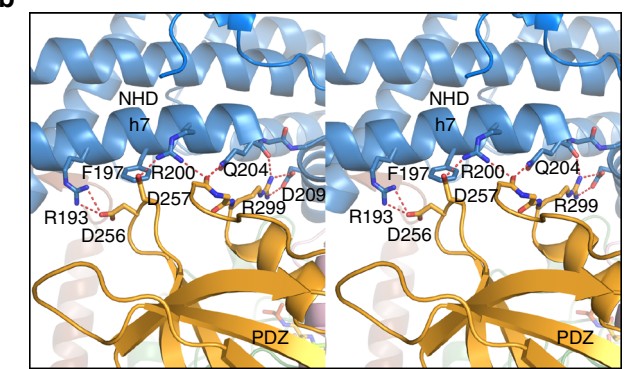

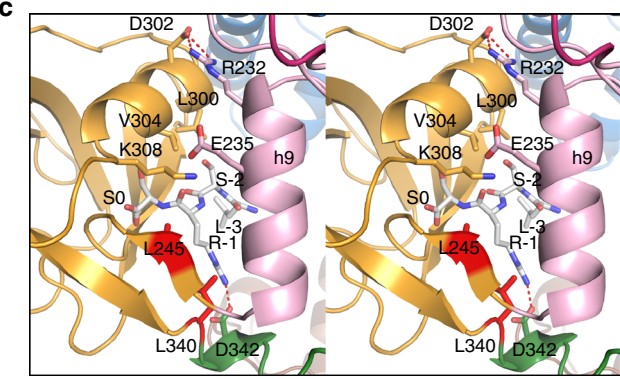

**Fig. 8** Large rotational movement of the PDZ domain and sensing of the bound PDZ ligand by the hinge residues in Prc. **a** Superimposition of the structures of two Prc molecules (chains C and D colored in green and orange, respectively), D1P (gray), and the resting form of CtpB (yellow), which reveals the hinge residues L340 and L245 and suggests a large rigid-body rotation of the PDZ domain of Prc during its proteolytic action. For clarity only the portions of the proteases connecting to the PDZ domains are shown, in two orthogonal views. The substrate passage perpendicular to the page plane is indicated by the black triangle. The r.m.s.d. between aligned Prc and CtpB (sequence identity: 26%) and between aligned Prc and D1P (sequence identity: 25.4%) are 3.09 Å (for 280 residues) and 2.78 Å (for 202 residues), respectively. **b** Stereo view of the interaction between the PDZ and NHD domains, colored in gold and blue ribbons, respectively, in Prc (chain C). **c** Stereo view of the hinge residues L245 and L340 interacting with the peptide ligand bound to the PDZ domain of Prc (chain C)

PDZ domain, which may exert effort/force by rotational movement (Fig. 9f). The substrate MepS is thought to dock into the cradle with its flexible C-terminal tail bound to the substrate-binding passage and the C-terminal end is captured by the resting PDZ domain from the opposite side. Binding of the substrate C terminus may induce large rotational movement of the PDZ domain in Prc, which is likely stabilized by interaction with the NHD and the hinge residues; the movement of the substrate C terminus-bound PDZ domain may drive the translocation of the

MepS polypeptide chain through the substrate-binding passage where the proteolytic site resides. Characterizing the details on how the lever-like features of the sNlpI-Prc complex are involved in degradation of MepS require further studies.

In conclusion, we have determined the structure of Prc, a monomeric PDZ protease forming a self-compartmentalized bowl-like structure with a hinged lid. The structure of the sNlpI-Prc complex and biophysical characterizations of their interaction provides molecular details on how a PDZ protease binds to a substrate adaptor with four TPRs. The lipoprotein NlpI has the dual role of bringing the soluble periplasmic protease Prc to the OM, where the lipid-anchored substrate MepS resides, and enhancing the efficiency of MepS degradation by Prc. This work also explains the structural and functional difference between the substrate-degrading and the C-terminal processing types of PDZ proteases. The sNlpI-Prc complex reveals lever-like structural features, which may be implicated in the degradation mechanism. The structure presented in this work may facilitate design of specific inhibitors targeting the proteolytic activity of Prc. Recent studies have suggested that the activity of Prc may contribute to the pathogenicity and virulence of several Gram-negative pathogens[17, 18]; suppressing Prc activity may prevent evasion of *E. coli* from complement-mediated serum killing[18]. Inhibiting the activity of Prc or its NlpI-interaction results in excessive cross-link hydrolysis and may affect the viability of Gram-negative bacteria. The possibilities for development of new antibacterial agents targeting the Prc-NlpI system await future proof-of-concept studies.

## Methods

**Cloning and mutagenesis**. DNA sequences encoding sNlpI (residues S20-Q294), sNlpI-ΔN (T37-Q294), and sMepS (C27-S188) without the signal peptides were cloned into pET28a vector encoding an N-terminal His-tag. sMepS (S28-Q294) was also cloned into pET21a with a C-terminal His-tag. Full-length Prc (M1-K683) with a C-terminal His-tag and untagged full-length NlpI were cloned into expression vectors as described previously[18, 19]. All NlpI and Prc mutants were generated by using PCR-based site-directed mutagenesis approach. The primer sequences used in this study are listed in Supplementary Table 1. All constructs were sequenced fully to confirm their identities.

**Expression and purification of proteins**. All recombinant proteins were expressed in *E. coli* BL21(DE3) cells (Invitrogen), except for Prc mutant proteins, including PrcΔPDZ (Δ244-339), Prc-K477A, Prc-K308W, Prc-L245A, Prc-L340G/A, and Prc-L340G/L245A, which were expressed in the Δprc strain MR812. Cells were cultured to an optical density at 600 nm of 0.6–0.8 and induced with 1 mM iso-propyl β-D-thiogalactopyranoside at 25 °C for 4 h. Cell pellets were suspended in lysis buffer containing 50 mM Tris-HCl, pH 8.0, 500 mM NaCl and then ruptured by French press (Avestin). After centrifugation at 35,000 × g at 4 °C for 45 min, the supernatant was applied to a nickel-nitrilotriacetic acid (Ni-NTA) agarose (Qiagen) column and washed with 20 mM imidazole. The protein fraction eluted with 250 mM imidazole was purified further by MonoQ 5/50 GL column (GE Healthcare) at pH 8.0, and Superdex 200 10/300 GL column (GE Healthcare) was equilibrated in 20 mM Tris-HCl, pH 8.0, 100 mM NaCl for Prc and sNlpI, but in 20 mM Tris-HCl, pH 7.5, 300 mM NaCl, and 1 mM DTT for sMepS. Protein purity was analyzed by SDS-PAGE.

**Crystallization and data collection**. The proteolytically inactive variant Prc-K477A was made for crystallization. Purified Prc-K477A was incubated with sNlpI at a molar ratio of 1:3 overnight and the complex was purified by Superdex 200. Crystals were obtained by mixing 1 μl of the protein solution concentrated to 8–10 mg ml⁻¹ with an equal amount of reservoir solution consisting of 0.2 M ammo-nium citrate, pH 7.4, and 14% PEG3350 using hanging-drop vapor diffusion method at 16 °C. The cubic crystals were cryoprotected by a brief transfer to the mother liquid supplemented with 20% glycerol before data collection. The crystal giving the best X-ray diffraction to 2.30 Å was used for data collection of a total of 220 frames with 0.5° oscillation per frame. The diffraction data was collected on NSRRC beamline 15 A (Taiwan). All images were indexed, integrated, and scaled using the HKL-2000 package[20]. Data collection statistics are reported in Table 1.

**Structure determination**. The structure was solved by molecular replacement using PHASER[2, 21]. The NlpI structure (PDB code 1XNF) and a poly-alanine model of the PDZ-missing protease core from CT441 (PDB code 4QL6), generated

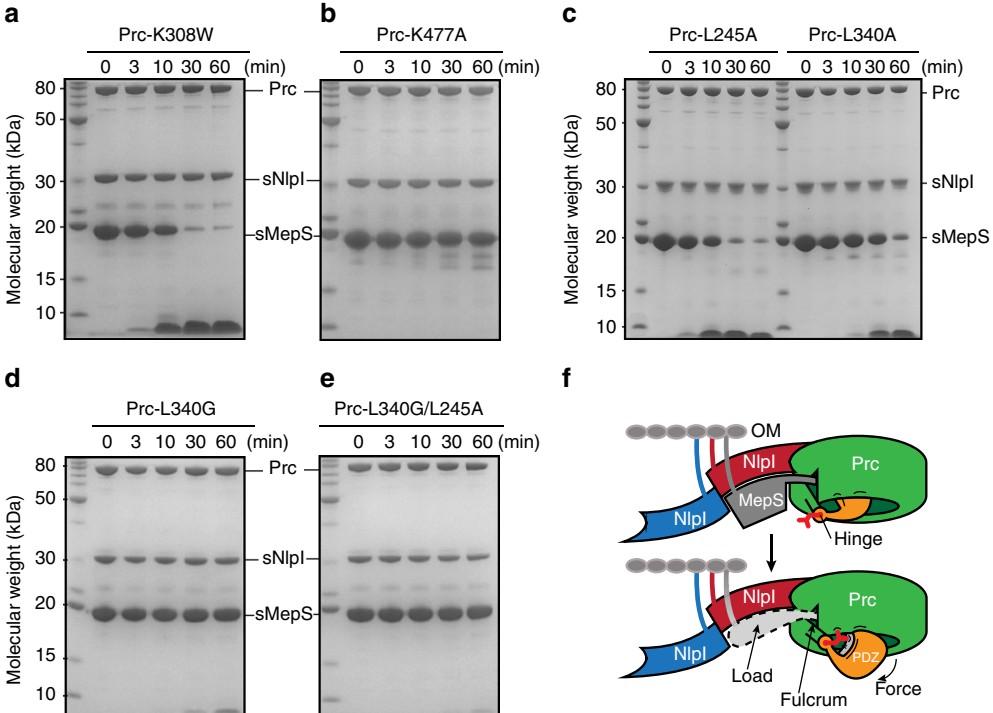

**Fig. 9** Role of the hinge residues of Prc in NlpI-mediated substrate degradation. **a–e** SDS-PAGE assays monitoring sNlpI-mediated sMepS degradation by Prc-K308W **a**, the catalytically inactive Prc-K477A as a reference **b**, Prc-L245A **c**, Prc-L340A **c**, Prc-L340G **d**, and Prc-L340G/L245A **e**. **f** A lever-like working model for degradation of protein substrate containing a flexible C-terminal tail by the Prc-NlpI complex. The initial cycle of substrate-docking (top) and ligand-triggered PDZ movement (bottom) steps are shown. See Discussion for details. The substrate C terminus is indicated by an asterisk. The ligand-sensing hydrophobic residues at the hinge are represented by a bifurcated side chain colored in red. The lipid anchors are depicted by colored curved lines linking to the outer membrane (OM). Sites for intramolecular interaction (e.g., the NHD and helix h9) and ligand binding in Prc are depicted by short black curves

by CHAINSAW[22], were used as the search models. Rigid-body refinement of the solution, in which one NlpI homodimer and the homologous protease domains from the two CT441 poly-alanine models were positioned correctly, gave an $R_{free}$ of 43.7%. Outside the protease domains, the initial electron-density map was only interpretable for some parts of the NTD, PDZ, and the CTD domains of Prc. Nevertheless, the map was subsequently improved by iterative cycles of extensive manual building of the protease, NTD, and PDZ domains of Prc in Coot[23], and refinement using Refmac5[24]. Then, with one cycle of autobuilding using Buccaneer[25], a 95% complete model with most of the CTD domains built was obtained with the $R_{work}$ and $R_{free}$ values of 25.5% and 32.0%, respectively. Finally, the structure was manually re-adjusted and water added. The final refined model, which contained residues 19–288 of NlpI and residues 25–673 of Prc, had $R_{work}$ and $R_{free}$ values of 21.8% and 26.5%, respectively (Table 1). All protein model figures were generated with PyMOL (v.1.7.2; Schrödinger). The atomic coordinates and structure factors for the Prc-sNlpI complex have been deposited in the Protein Data Bank (http://www.rcsb.org/) with the accession number 5WQL.

**Docking analysis.** Docking of MepS (PDB code 2K1G) to the crystal structure of the sNlpI-Prc complex (this work) was initially carried out using the ClusPro 2.0 protein-protein docking server (http://cluspro.bu.edu)[26]. No residue constraints were supplied as inputs for docking calculation. The sNlpI-Prc complex was assigned as the receptor in the docking calculation. Two of the top 10 docking models predicted by ClusPro placed MepS in a prominent valley formed between the NlpI homodimer and Prc in the complex. Although the globular MepS in the valley were docked in different orientations, the revealed putative MepS-docking valley appeared convincing because, after manually performing rigid-body rotation, MepS was re-oriented such that its lipid-anchored N-terminal coil region is well accommodated in the complex structure while allowing its flexible C-terminal tail positioned precisely at the entrance of the substrate-binding passage of Prc, as shown in Fig. 6.

**Analytical ultracentrifugation.** AUC experiments were carried out using an XL-A AUC with An-50 Ti rotor (Beckman Coulter). Sedimentation velocity measurements using absorbance optics of reference buffer and samples in 20 mM Tris-HCl, pH 8.0, 100 mM NaCl, 2 mM DTT were performed at 20 °C at a speed of 163,000 × g. The buffer density and viscosity were calculated using Sednterp and data were

analyzed with the standard c(s) model in Sedfit and plotted using GraphPad Prism (GraphPad Software, USA)[27, 28]. Experiment was repeated two times with similar observations, and representative results are shown.

**SEC-MALS analysis.** SEC-MALS measurements were carried out with a mini-DAWN TREOS detector (Wyatt Technology Corporation) coupled to an Agilent 1260 Infinity HPLC. A total of 240–360 μg protein samples were injected into a size exclusion chromatography column (ENrich SEC 650, Bio-Rad) and continuously run at a flow rate of 0.5 ml min$^{-1}$ in the buffer containing 20 mM Tris-HCl, pH 7.5, 200 mM NaCl, 2 mM DTT, and 0.02% NaN$_3$. The molecular weight was determined by multi-angle laser light scattering using an in-line miniDAWN TREOS detector and an Optilab T-rEX differential refractive index detector (Wyatt Technology Corporation). Bovine serum albumin (Sigma, A1900) was used for system calibration and the data were analyzed using ASTRA 6 Software (Wyatt Technology Corporation) with the dn/dc value set to 0.185 ml g$^{-1}$. Experiment was repeated two times with similar observations, and a representative figure is shown.

**Isothermal titration calorimetry.** Calorimetric titrations of sMepS with sNlpI were performed on an iTC200 microcalorimeter (MicroCal) at 20 °C. Protein samples were extensively dialyzed against a buffer containing 20 mM HEPES, pH 7.5, 300 mM NaCl and 2 mM 2-mercaptoethanol. All solutions were filtered using membrane filters (pore size 0.20 μm and thoroughly degassed for 10 min by sonicator). The sample cell (200 μl) and the injection syringe (40 μl) were filled with 23 μM of sMepS or Prc solutions and 230 μM of the titrating WT or mutant sNlpI proteins (alone or mixed with an equal molar ratio of Prc), respectively. For each titration experiment, a preliminary 0.2-μl injection was followed by 15 subsequent 2.49-μl injections. Binding isotherms were calculated by plotting the integrated heat peaks, normalized by the moles of injectant, against the molar ratio of total injectant to total protein in the cell at each injection. The data were fitted to one binding site model using Origin 7.0 (MicroCal). Experiments were repeated at least three times, and representative results are shown.

**Ni-NTA pull-down assay.** Equal total protein amount of cleared cell lysates of His$_6$-tagged WT or mutants of sNlpI was mixed with that of untagged Prc-K477A in binding buffer containing 50 mM Tris-HCl, pH 8.0, 200 mM NaCl for 2 h on ice, followed by incubation with 1 ml Ni-NTA resin (Qiagen) by end-to-end rotation

for 1.5 h at 4 °C. The resin was collected by centrifugation at $500 \times g$ for 10 min and then washed twice with 5 ml binding buffer containing 20 mM imidazole. The bound protein was eluted with the binding buffer containing 250 mM imidazole. Samples were analyzed by SDS-PAGE. Experiment was repeated two times with similar observations, and a representative figure is shown.

**Degradation assays by SDS-PAGE**. Each reaction was composed of 7 μg sMepS incubated with the complex of 1 μg WT or mutants of sNlpI plus 2 μg WT or mutants of Prc, in a molar ratio of 1:1, in a reaction mixture (16 μl) containing 20 mM Tris-HCl, pH 8.0, 150 mM NaCl, and 2 mM DTT at 37 °C. At different time points, the reactions were stopped by adding 5 × SDS-PAGE loading dye and heating at 95 °C for 5 min. The samples were then loaded onto a NuPAGE gel (4–12% Bis-Tris) (Invitrogen). Substrate protein bands were detected by Coomassie blue staining. For peptide inhibition assays, the C-terminal peptide of MepS: NH2-RYNEARRVLSRS-COOH, at a 0.25, 0.5, 1, or 2 mM concentration, was incubated with Prc for 15 min prior to adding the protein substrate. Experiments were repeated three times with similar results, and representative figures are shown.

For determination of the specific MepS degradation activities of Prc, 4 μg of MepS was incubated with various amounts of Prc (from 0.25 to 2 μg) in the absence or presence of 1:1 molar ratio of WT-NlpI or NlpI-QM (D113A/E117A/D120/E124A) for either 3 min (with WT-NlpI) or 1 h (Prc alone or with NlpI-QM) at 37 °C. All reactions were analyzed by SDS-PAGE; each gel was also loaded with three known amounts of MepS (from 0.5 to 4 μg) as internal controls. After staining with Coomassie blue, gel images were scanned and the bands corresponding to full-length MepS were analyzed by Quantity One 1-D Analysis Software (Bio-Rad). Experiments were repeated three times with similar results.

**In vivo viability assay**. Overnight grown cultures were serially diluted in minimal media and 5 μl of each dilution was spotted on the indicated plates and incubated at appropriate temperature overnight. The plates contain 1.5% agar in either LB (1% tryptone, 0.5% yeast extract, 1% NaCl), LBON (1% tryptone, 0.5% yeast extract), or nutrient broth (NA; 0.5% peptone and 0.3% beef extract). The experiments were repeated at least two times with similar observations, and representative results are shown.

**Differential scanning fluorimetry**. Five micromoles of 5 mM NlpI, Prc, NlpI-Prc, and NlpI-QM-Prc proteins were mixed with SYPRO Orange (Sigma-Aldrich) in 20 mM HEPES, pH 7, 150 mM NaCl. Mixtures were transferred to the LightCycler 480 Multiwell plate 96 (Roche) and sealed using LightCycler 480 Sealing Foil (Roche). Protein melting experiments were performed in a real-time PCR machine (LightCycler 480II, Roche), which was set up with a detection format of 465 nm as the excitation wavelength and 580 nm as the emission wavelength to detect Sypro Orange fluorescent signal. Fluorescence in function of the temperature was recorded in a temperature range from 20 °C to 95 °C at $0.01 \, °C \, s^{-1}$ rate. The $T_m$ was determined as the minimum of negative first derivative curves calculated from the melting curves and plotted using Prism6 (GraphPad Software). The experiments were repeated two times with similar observations, and a representative is shown.

**Data availability**. The atomic coordinates and structure factors have been deposited in the Protein Data Bank, www.pdb.org, with accession code 5WQL. Other relevant data are available from the corresponding author upon reasonable request.

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

## Acknowledgments

We thank Chia-Cheng Feng for crystallizing the sNlpI-Prc complex, Chiao-I Kuo for technical assistance, and the beamline support from NSRRC, a national user facility supported by the Ministry of Science and Technology, Taiwan, ROC. This work was supported by funds from Council of Scientific and Industrial Research and Department of Biotechnology, Government of India (to M.R.), and from Academia Sinica and Ministry of Science and Technology (under Grant MOST105-2320-B-001-015-MY3), Taiwan, ROC (to C.I.C.).

## Author contributions

M.Y.S. and C.I.C. conceived the study. C.H.T. and D.M.L. provided materials and reagents. M.Y.S., N.S., S.C.S., M.R.H., S.R.T., M.R. and C.I.C. designed the

experiments and analyzed the data. M.Y.S. solved the crystal structure. M.Y.S., N.S., C.Y. W., S.C.S., Y.T.K., L.C.K., M.R.H. and S.R.T. performed the experiments. M.Y.S. and C.I.C. wrote the manuscript with inputs from all other authors.

## Additional information

**Competing interests:** The authors declare no competing financial interests.

