## [Peer Review file · Nature Communications]

Reviewers' comments:

Reviewer #1 (Remarks to the Author):

The manuscript by Mig-Yuan Su et al present a potentially very interesting structure explaining the regulation of peptidoglycan hydrolysis in E.coli (the actual function is not very specifically written out in the title – the title is bit too generic perhaps?).

The structure presented is of high value and deserving of a visible journal for publication, if not here in some other high impact journal after proper revision and I would like to congratulate the authors on solving the complex structure.

However, there are some short-comings that might compromise the suitability of the paper for Nature communications as the authors do not present the impact very clearly and text is quite descriptive/focused on the mechanism and not so much emphasis is put on clearly explaining the impact and all data is not clearly presented that is needed to back up the model – also conclusions at the end seem to be missing altogether, which appears bit odd. The paper requires rewriting in these aspects before its acceptable in format, while overall I believe the results could lead for publication in current journal with corrections.

More specific comments:

Abstract does not well reflect what is the major impact (regulation of PG degradation) of this paper and should be re-written completely. The proteolytic and PDZ binding sites are not clearly explained – how are these related should be obvious from the context of the abstract text.

Fig 1.: Fig 1a the corresponding molecular weight estimates should be given with precision estimate (please do not omit actual data).

Fig1b – Mw (230Kda) again should be given more exactly based of the SEC-MALS with error reported. – typically RI is given as the signal not the LS – MW is calculated with RI as concentration detector – please update to either UV or RI trace.

Structures: please mention overall dimensions of the complex and molecular weights of each protein and e.g. the number of amino acids residues present in the structural models (this is buried in the methods and for Prc not even there?). Overall size and exact content of each construct should be given.

Fig2D is too small for stereo (ideal distance between stereo figures is 6 cm) and labels are hardly visible please make them bigger (this might be a formatting issue of the manuscript version also). All discussed residues and helices etc must be also visible in the figures to the eye.

Also the co-crystallized peptides are completely invisible please remake the figure such that it is actually readable. What in Fig 2b is the lipoprotein substrate access (indicated) based on? This is not discussed. Remove or explain.

Activity of Prc – the enzyme to substrate ratio in Fig1 is 1:1, this seems high? How active is the protein? I would like to see kinetics for proteolysis either referred to or measured to verify that the protein is properly active. As it stands its very descriptive and from the gels it looks like excessive amounts of enzyme and co-factor (Nlpi) are needed for substrate degradation to occur. This raises the question if the proteins are fully active. Please provide data on the kinetics of this system either a literature reference or a small assay.

p.6 lines 89-90 description is confusing - presumably "the dimer interface" refers to Nlpi? Not mentioned.

p.7 lines107-110 "open substrate binding passage is located.." what is this suggestion based on? These kinds of statements should be justified.

Line 117 – electron density map figures are not good/clear enough to show whether the modeling of the sequence was actually justified – please provided figures that are clearer and show the fit of the side chains.

p.8: how do the authors know actually where the proteolytic site is? No information is given on this. Please justify. I am concerned about what conclusions can be drawn based on the data presented - are the conserved residues or mutagenesis data in the literature verifying the active sites? If not the suggested sites must be verified by mutagenesis on Prc.

p.11 TM and QM mutants not explains (I assume triple and quadruple – what is the triple?)

Figure 5 and associated text – how the docked model was generated must be properly explained in methods. How is this validated? Also, labels in Fig 5. are too small and the figure is too full. The red spheres are not visible – should be omitted for clarity?

Please redo the figure. Figure on top-right is not labeled PDZ domain (?) in the background could be removed for clarity. Helices and b-strands labeled are impossible to locate in Fig 5a. (right). ITC: what is the affinity of the substrate for Prc (if K_m is not possible to derive)? How is the complex formed? in which order? For this it would be good to know the affinities for different combinations of proteins/or something about the kinetics of the system. Could the authors provide affinities for different interactions or explain why only the affinity for the NlpI was measured?

Please explain structural similarity between Prc and CtpB and D1P (r.m.s.d. of alignment and for how many residues?). Again Figure 6 images should be made bit larger to be readable. The structure figures are the main content and should be made clearly interpretable and supported accordingly in the text.

p.18 TPR protein complexes – at least the anaphase promoting complex contains several TPR subunits in one large complex (as shown by Barford and colleagues).

Line 243 "helical interaction" should be helix-helix interaction?

p.19 line 250 "By associating..." How do helices extend a beta-sheet? I don't understand this sentence. The conclusion is also bit speculative – How does NlpI form a "substrate docking scaffold" here exactly or what does this mean?

Line 257 – what is meant by "resistance"? Please elaborate.

The text should end with some kind of conclusion on the impact – without this its hard to see the general impact of the paper.

I hope the comments will be helpful.

Reviewer #2 (Remarks to the Author):

The manuscript by Su et al. presents the structure of a tetrameric protein complex consisting of the PDZ protease Prc and the stabilizing lipoprotein NlpI from E.coli. The authors used soluble variants of the membrane anchored proteins to investigate their function in SDS-PAGE degradation assays and in in vitro viability assays. Protein interactions of NlpI and the protease substrate MepS were verified using isothermal titration calorimetry. The structure of the soluble protein complex consisting of 2 molecules of a catalytically inactive version of Prc (K477A) and NlpI was solved by molecular replacement. Together with previous structural evidence from a related PDZ protease,

the authors use this data to propose a mechanism for the degradation of lipoproteins by the Prc-NlpI complex.

Overall, the manuscript is well written, however, the authors should elaborate on why this publication is of general interest to the readership of Nature Communications. There are also a number of major issues that have to be addressed before the manuscript is suited for publication in this journal. Please find a list of the major and minor issues below.

Major points:

1) The abstract should address a more general readership. A one-sentence explanation of the target is followed by a very detailed description of the results, which is very confusing at this point. It is suggested that the authors should elaborate on the function and biological significance of the target complex.

The biological significance and broader interest of this study should also be more thoroughly discussed in the main text of the manuscript (e.g. is this complex a target for antibiotic development?).

2) The authors state that sMepS was inefficiently degraded by Prc, due to instability of the latter. The authors demonstrate that addition of sNlpI, but not sNlpI-QM, to Prc results in a more efficient degradation of the substrate. However, the effect of NlpI on Prc is not clearly established.

To what extent is the stability of Prc enhanced? Could it be that NlpI has an effect on the catalytic activity rather than the stability of Prc?

The author's statement would be supported by measuring the thermal stability the Prc alone and in presence of NlpI-QM or in complex with NlpI.

In addition, the authors should further try to quantify the catalytic activity of Prc alone and in complex with NlpI.

3) According to the Methods section, the authors crystallized and solved the structure of a catalytically inactive Prc mutant (K477A). However, this important fact is not mentioned anywhere in the main text of the manuscript.

4) The authors compare the binding pockets of the proteolytic and the PDZ ligand binding sites (line 120 ff.). They come to the conclusion that the matching substrate/ligand binding pockets in Prc, which is in contrast to CtpB, suggest that its PDZ domain binds the nascent C-term after cleavage and provide the superposition of the pockets in Fig. S3 as evidence. It is very hard to read Fig. S3, which substantially weakens this statement. In addition, a mere overlay of the binding pockets and comparison with CtpB is not ideally suited to support this finding. The authors could try to use the peptide fragment LSRS-COOH, observed in one of the PDZ domains, as a model and dock it into the other binding site.

5) line 160 ff: How was the model of the Prc-NlpI-MepS model exactly constructed? It is not mentioned in the methods section. Furthermore, to validate the model, the authors mutated L38A/C and R82E. Those mutations are suggested to abrogate the interaction between MepS and sNlpI as indicated by the ITC experiments. However, the authors also state that monomeric sNlpI does not interact with MepS. Therefore, the authors need to provide evidence that the mutations L38A/C and R82E affect the direct interaction of MepS and NlpI and not merely prevent dimerization of NlpI.

The model suggests that MepS also interacts with Prc, that then mediates the important steps in the degradation of the substrate. It is therefore not clear, why the authors did not include Prc in these experiments.

6) line 203 ff: The authors provide evidence that the Prc mutant L340G has an impaired activity and that the double mutant Prc-L340G/L245A almost lost activity. It would therefore be of interest to investigate the effect of the Prc-L245A mutant alone as well, since the difference in the activities between Fig. 6e and f seems small.

Furthermore, can the authors comment on the fact that the two conserved residues L340 and L245

were mutated to G and A, respectively? Was the L340A mutant still active? It would make sense to mutate both leucines to the same amino acid.

7) It makes sense to assume that the PDZ domain can adopt a closed conformation similar to the PDZ resting state in CtpB. However, the model that was constructed manually in Fig S5b is not useful in supporting this finding. It is no proof of this assumption and one can hardly read anything from this Figure.

For reasons of comparison, it would also be interesting to know the dimensions of the movements (in angstrom) of the PDZ domains in CtpB and Prc (D1P).

8) Is there evidence in the literature to support the hypothesis that the large movement of the PDZ domain, along with the unique structure of Prc may provide a mechanical pulling force for the degradation of substrate (lines 233-237)?

9) The authors come up with a hypothesis for the "leverage" mechanism for substrate-degradation (line 254 ff.). Based on the evidence provided, however, it is hard to understand the rational basis of these assumptions.

Specifically, what is the structural basis for "substrate-deformation" (line 263) or that the substrate peptide chain is "pulled by rotational movement of the PDZ domain" (line 264) that allows cleavage by the catalytic dyad. These ideas haven't been adequately developed anywhere in the manuscript, nor is there any evidence from literature provided that supports this mechanism. In my opinion, these assumption would require co-crystal structures of the complex with substrates in different states and/or extensive molecular dynamics simulations. Hence, I think it is too overstated to include in the manuscript.

Minor points:

line 26: Abbreviation of out membrane (OM) is not defined.

Line 26: Abbreviation of TRP is not defined.

line 65-66: The authors state that MepS and NlpI were expressed as soluble forms without their signal peptides. The methods state that NlpI is truncated before S20. It is not clear if the lipoprotein's invariant cysteine is still part of the construct, which could lead to posttranslational modification and membrane anchoring.

line 73-75: The authors state that reduced lysozyme is expected to have an unstructured C-terminus. Is there evidence of this in the literature or do the authors assume this?

line 114-115: It is not mentioned if the peptides that co-crystallized were added by the authors or co-purified.

Would it be possible to identify the peptide in the proteolytic groove and in the ligand binding site by mass spectrometry?

line 141 ff: How does the activity of Prc in presence of sNlpI-QM compare to the activity of Prc alone? Quantification of the catalytic activity would be useful to support and to clarify these findings (see major point 2).

line 160 ff: How was the model of the Prc-NlpI-MepS model constructed and validated? It is not mentioned in the methods section.

line 193: "h7 of the NTD"... should this read NHD?

line 202: "Lke the wild type"... should read "Like the wild type"

line 214: "structure of substrate-degrading Prc" is overstated as a catalytically inactive mutant was

crystallized.

line 245: Is there anything known about the specific functions of NlpI, apart from its ability to bind to Prc and to “acts as a hub to coordinate the assembly of multi-protein complexes” as presented here? The authors should elaborate on this.

Reviewers' comments:

Reviewer #1 (Remarks to the Author):

The manuscript by Ming-Yuan Su et al present a potentially very interesting structure explaining the regulation of peptidoglycan hydrolysis in E.coli (the actual function is not very specifically written out in the title – the title is bit too generic perhaps?).

The structure presented is of high value and deserving of a visible journal for publication, if not here in some other high impact journal after proper revision and I would like to congratulate the authors on solving the complex structure.

However, there are some short-comings that might compromise the suitability of the paper for Nature communications as the authors do not present the impact very clearly and text is quite descriptive/focused on the mechanism and not so much emphasis is put on clearly explaining the impact and all data is not clearly presented that is needed to back up the model – also conclusions at the end seem to be missing altogether, which appears bit odd. The paper requires rewriting in these aspects before its acceptable in format, while overall I believe the results could lead for publication in current journal with corrections.

We are grateful to the reviewer #1 for all the valuable comments and suggestions allowing us to improve the manuscript. We have now made a more specific title “Structural basis for NlpI-dependent degradation of MepS by the PDZ protease Prc” for the paper. We have modified the introduction and Discussion, which now include more on the biological impact regarding the complex; conclusions have been added to the last paragraph. Also we have preformed all suggested experiments and the results have been included in the revised manuscript (also see our point-by-point response below).

More specific comments:

Abstract does not well reflect what is the major impact (regulation of PG degradation) of this paper and should be re-written completely. The proteolytic and PDZ binding sites are not clearly explained – how are these related should be obvious from the context of the abstract text.

We have rewritten the abstract, which now highlights the major impact, with a self-contained sentence regarding the proteolytic and PDZ binding sites.

Fig 1.: Fig 1a the corresponding molecular weight estimates should be given with precision estimate (please do not omit actual data).

We have described in Results the Mw with precision estimate calculated by AUC.

Fig1b – Mw (230Kda) again should be given more exactly based of the SEC-MALS with error reported. – typically RI is given as the signal not the LS – MW is calculated with RI as concentration detector – please update to either UV or RI trace.

We have added to the figure and described in Results the Mw with precision estimate calculated by SEC-MALS and updated the figure plot using the UV trace.

Structures: please mention overall dimensions of the complex and molecular weights of each protein and e.g. the number of amino acids residues present in the structural models (this is buried in the methods and for Prc not even there?). Overall size and exact content of each construct should be given.

We have now mentioned overall dimensions of the complex and the molecular weight of the each protein in Results. The number of amino acid residues present in the structure and specific details about the constructs have now been updated in Methods.

Fig2D is too small for stereo (ideal distance between stereo figures is 6 cm) and labels are hardly visible please make them bigger (this might be a formatting issue of the manuscript version also). All discussed residues and helices etc must be also visible in the figures to the eye.

We have enlarged the stereo figure, which is now in Fig. 3b; the distance between the centers of the two views is now ~6 cm.

Also the co-crystallized peptides are completely invisible please remake the figure such that it is actually readable. What in Fig 2b is the lipoprotein substrate access (indicated) based on? This is not discussed. Remove or explain.

We have enlarged the Fig. 2b and labeled the four co-crystallized peptides shown in rainbow-colored spheres. And we have removed the labels for the lipoprotein substrate access sites from the figure.

Activity of Prc – the enzyme to substrate ratio in Fig1 is 1:1, this seems high? How active is the protein? I would like to see kinetics for proteolysis either referred to or measured to verify that the protein is properly active. As it stands its very descriptive and from the gels it looks like excessive amounts of enzyme and co-factor (Nlpi) are needed for substrate degradation to occur. This raises the question if the proteins are fully active. Please provide data on the kinetics of this system either a literature reference or a small assay.

We thank for reviewer for the comments. Prc was expressed and purified based on earlier studies (1992-1996) by Robert Sauer's lab, which has well characterized the enzymatic activity of Prc using small fluorogenic peptide substrates, cited in our manuscript. In the present work we instead focused on Prc's degradation activity on intact protein substrate, ie. MepS, in the absence or presence of Nlpi. For all time course assays, a reaction mixture containing 7 ug of sMepS, 2 ug of Prc, and 1 ug of sNlpi were used in each well of the SDS-PAGE gel, actually corresponding to a molar ratio of sMepS:Prc:sNlpi = 14:1:1. Our results are consistent with previous gel analysis of the reaction using the similar molar ratio published in PNAS by Sigh *et al.* (2015), which is also cited in our manuscript. We understand that enzymatic activity is normally carried out with >100~1000 fold excess of substrate. However, we performed MepS degradation assays in the above condition based on two considerations: (1) to avoid overloading the polyacrylamide gel; (2) we thought that since MepS and Nlpi are both lipid-anchored *in vivo*, presumably with limited diffusibility, soluble Prc must first bound to Nlpi in order to reach MepS on the outer membrane (OM), the related amounts of MepS versus Nlpi-Prc may be stoichiometric.

Nevertheless, per the reviewer's suggestion we have now determined the specific sMepS degradation activities monitoring the disappearance of a fix amount of full-length MepS on gel using serial dilutions of Prc alone or with Nlpi in 1:1 molar ratio; the results have been included in the revised manuscript in Results and Methods. The kinetic assay showed that the specific activities of total sMepS degradation (by multiple cleavages against each molecule of sMepS) were 0.580 ± 0.102 nmole/min/mg

without NlpI and 27.445 ± 6.800 nmole/min/mg with NlpI, which suggests that sNlpI enhances the sMepS degradation activity of Prc by ~50-fold. Note that these activity values may be underestimated because lipid-anchored MepS and NlpI, with a relatively fixed orientation on the OM than the soluble forms used in the assays, may further promote a more productive encounter with Prc.

p.6 lines 89-90 description is confusing - presumably “the dimer interface” refers to NlpI? Not mentioned.

The reviewer is right and the term has been updated as “the NlpI dimer interface”.

p.7 lines 107-110 “open substrate binding passage is located..” what is this suggestion based on? These kinds of statements should be justified.

The description is based on its structural similarity with the active structure of CtpB. Moreover, a co-crystallized peptide is bound inside the conserved passage. We have modified the sentence accordingly and cited the reference therein.

Line 117 – electron density map figures are not good/clear enough to show whether the modeling of the sequence was actually justified – please provide figures that are clearer and show the fit of the side chains.

We have re-made the map figures in Fig. S2 according to the suggestion to show the fit of the peptide side chains.

p.8: how do the authors know actually where the proteolytic site is? No information is given on this. Please justify. I am concerned about what conclusions can be drawn based on the data presented - are the conserved residues or mutagenesis data in the literature verifying the active sites? If not the suggested sites must be verified by mutagenesis on Prc.

The catalytic site consisting of the S452-Lys477 dyad and S1-2 binding pockets have been defined based on previous biochemical studies by Robert Sauer’s lab and the structural study of CtpB by Mastiny *et al.*, the papers have been cited in the manuscript. Our mutational study of Prc-K477A also confirmed the critical role of K477 in the

proteolytic activity (Fig. 9b).

p.11 TM and QM mutants not explains (I assume triple and quadruple – what is the triple?)

The mutations of the TM and QM mutants had been specified in the figure legends. We have now explained the TM and QM mutants in the main text also.

Figure 5 and associated text – how the docked model was generated must be properly explained in methods. How is this validated?

We thank the reviewer for the questions. First of all, our docked model was validated first by its consistency with previous finding that Prc recognizes the C-terminus of protein substrates, and further by site-directed mutagenesis as presented in the sub-section entitled “*TPRI of NlpI is involved in MepS binding*”. Per the reviewer’s request, docking analysis has now been described in Methods. Basically, MepS (PDB code 2K1G) was first docked to the crystal structure of sNlpI-Prc complex using the ClusPro 2.0 protein-protein docking server (<http://cluspro.bu.edu>) (Kozakov et al., 2010). Two of the top 10 docking models predicted by ClusPro placed MepS in a prominent valley formed between the NlpI homodimer and Prc in the complex. Although the globular MepS in the valley were docked in different orientations, the revealed putative MepS-docking valley appeared convincing after manually performing rigid-body rotation, MepS was oriented such that its lipid-anchored N-terminal coil region is well accommodated in the complex structure while allowing its flexible C-terminus positioned precisely at the entrance of the substrate-binding passage of Prc (Figs. 6a,b).

Also, labels in Fig 5. are too small and the figure is too full. The red spheres are not visible – should be omitted for clarity? Please redo the figure. Figure on top-right is not labeled PDZ domain (?) in the background could be removed for clarity. Helices and b-strands labeled are impossible to locate in Fig 5a. (right).

We have now enlarged the images (now in Fig. 6) to make the figures and labels more visible. The figures serve primarily to show the MepS-docking valley in the

overall structure, the structural elements flanking the valley, and the locations of the N- and C-termini of docked MepS.

ITC: what is the affinity of the substrate for Prc (if K_m is not possible to derive)? How is the complex formed? in which order? For this it would be good to know the affinities for different combinations of proteins/or something about the kinetics of the system. Could the authors provide affinities for different interactions or explain why only the affinity for the NlpI was measured?

In response to the reviewer's questions, we have performed the suggested ITC experiments and the results have been updated in Fig. 6 and in the main text in the sub-section entitled "*TPRI of NlpI is involved in MepS binding*". Consistent with previous pull-down results by Singh *et al.* (PNAS, 2015), it was found that Prc binds to NlpI with very high affinity ($K_d < 10$ nM); the value could not be precisely determined by ITC. The calculated K_d values for sMepS binding to sNlpI alone and the sNlpI-Prc complex were 0.75 and 2.09 μ M, respectively. These comparable K_d values suggest that the molecular surface of Prc in the NlpI-Prc complex may contribute little to the binding energy of MepS interaction. Accordingly, our ITC result confirmed no apparent binding of sMepS to Prc alone. Hence, we have focused on characterizing the interaction of MepS with the various NlpI mutants.

These data, together with the structural and biochemical results, suggest the following scenarios: (1) Prc by itself does not bind (or bind very weakly) to the lipid-anchored substrate MepS. (2) However, Prc binds to the lipid-anchored adaptor NlpI with a high affinity, forming a hetero-tetrameric complex (the crystal structure), which then allows MepS to bind (the docking model) and be degraded. (3) Alternatively, the lipid-anchored adaptor NlpI could form a complex with lipid-anchored MepS first. However, it has been shown that the cellular levels of both NlpI and Prc are constitutive; only the level of MepS fluctuates (Singh *et al.*, 2015). Therefore, pre-formation an NlpI-MepS complex for recruiting Prc may not be a biologically relevant process.

Please explain structural similarity between Prc and CtpB and D1P (r.m.s.d. of alignment and for how many residues?). Again Figure 6 images should be made bit larger to be readable. The structure figures are the main content and should be made

clearly interpretable and supported accordingly in the text.

The r.m.s.d. between aligned Prc and CtpB (sequence identity: 26%) is 3.09 Å (for 280 residues); the r.m.s.d. between aligned Prc and D1P (sequence identity: 25.4%) is 2.78 Å (for 202 residues), which are included in the figure legend.

We have now enlarged the images (now in Fig. 8) to make them more readable.

p.18 TPR protein complexes – at least the anaphase promoting complex contains several TPR subunits in one large complex (as shown by Barford and colleagues).

We thank the reviewer for the 2014 paper we missed, and we have removed the statement according to the fact.

Line 243 “helical interaction” should be helix-helix interaction?

Agreed; and we have changed the term accordingly.

p.19 line 250 “By associating...” How do helices extend a beta-sheet? I don’t understand this sentence. The conclusion is also bit speculative – How does NlpI form a “substrate docking scaffold” here exactly or what does this mean?

With this sentence we meant to suggest that, by associating with Prc, the NlpI homodimer contributes an array of helices, effectively extending the beta-sheet surface of the vault of Prc to form a substrate-docking cradle. We have modified the sentence as such and hope it makes sense now.

Line 257 – what is meant by “resistance”? Please elaborate.

By “resistance” we actually meant the resistance arm in a lever. Thanks to the reviewer, we have now modified the paragraph discussing the proposed working mechanism to avoid confusion.

The text should end with some kind of conclusion on the impact – without this its hard to see the general impact of the paper.

Thanks to the reviewer, we have now added a conclusion paragraph to the end of the main text discussing the impact of our study.

I hope the comments will be helpful.

Reviewer #2 (Remarks to the Author):

The manuscript by Su et al. presents the structure of a tetrameric protein complex consisting of the PDZ protease Prc and the stabilizing lipoprotein NlpI from E.coli. The authors used soluble variants of the membrane anchored proteins to investigate their function in SDS-PAGE degradation assays and in in vitro viability assays. Protein interactions of NlpI and the protease substrate MepS were verified using isothermal titration calorimetry. The structure of the soluble protein complex consisting of 2 molecules of a catalytically inactive version of Prc (K477A) and NlpI was solved by molecular replacement. Together with previous structural evidence from a related PDZ protease, the authors use this data to propose a mechanism for the degradation of lipoproteins by the Prc-NlpI complex.

Overall, the manuscript is well written, however, the authors should elaborate on why this publication is of general interest to the readership of Nature Communications. There are also a number of major issues that have to be addressed before the manuscript is suited for publication in this journal. Please find a list of the major and minor issues below.

Major points:

1) The abstract should address a more general readership. A one-sentence explanation of the target is followed by a very detailed description of the results, which is very confusing at this point. It is suggested that the authors should elaborate on the function and biological significance of the target complex.

The biological significance and broader interest of this study should also be more thoroughly discussed in the main text of the manuscript (e.g. is this complex a target for antibiotic development?).

We appreciate the reviewer #2 for all the insightful comments and suggestions on improving the work. Focusing more on the biological significance and impact of the

study, the Abstract have been re-written; the introduction and the last paragraph of Discussion have been modified based on the suggestion (changes highlighted in blue).

2) The authors state that sMepS was inefficiently degraded by Prc, due to instability of the latter. The authors demonstrate that addition of sNlpI, but not sNlpI-QM, to Prc results in a more efficient degradation of the substrate. However, the effect of NlpI on Prc is not clearly established.

To what extent is the stability of Prc enhanced? Could it be that NlpI has an effect on the catalytic activity rather than the stability of Prc?

The author's statement would be supported by measuring the thermal stability the Prc alone and in presence of NlpI-QM or in complex with NlpI.

In addition, the authors should further try to quantify the catalytic activity of Prc alone and in complex with NlpI.

We thank the reviewer for the good questions. Based on the suggestion, we have now included results of the thermal shift assay in Fig. 1e and showed that NlpI increase the thermal stability of Prc by forming a complex; NlpI-QM, which does not bind to Prc, fails to induce an increased T_m shift as wild-type NlpI does.

Per the reviewer's suggestion, we have also performed kinetic assays to determine the specific activity of Prc on total MepS degradation, as described now in the first sub-section of Results. The specific sMepS degradation activities of Prc alone and with sNlpI in 1:1 molar ratio were calculated to be 0.580 ± 0.102 and 27.445 ± 6.800 nmole/min/mg, respectively (see Methods), indicating a 50-fold enhancement of the specific activity mediated by sNlpI.

3) According to the Methods section, the authors crystallized and solved the structure of a catalytically inactive Prc mutant (K477A). However, this important fact is not mentioned anywhere in the main text of the manuscript.

We have specified the use of the mutant Prc-K477A for crystallography in the main text (in the 2nd sub-section on the overall structure).

4) The authors compare the binding pockets of the proteolytic and the PDZ ligand binding sites (line 120 ff.). They come to the conclusion that the matching substrate/ligand binding pockets in Prc, which is in contrast to CtpB, suggest that its

PDZ domain binds the nascent C-term after cleavage and provide the superposition of the pockets in Fig. S3 as evidence. It is very hard to read Fig. S3, which substantially weakens this statement. In addition, a mere overlay of the binding pockets and comparison with CtpB is not ideally suited to support this finding. The authors could try to use the peptide fragment LSRS-COOH, observed in one of the PDZ domains, as a model and dock it into the other binding site.

We thank the reviewer for the good idea! We have re-made Fig. S3 accordingly by using the peptide fragment LSRS-COOH as a model and dock it into the other binding site by superposition with the bound substrate peptide.

5) line 160 ff: How was the model of the Prc-NlpI-MepS model exactly constructed? It is not mentioned in the methods section. Furthermore, to validate the model, the authors mutated L38A/C and R82E. Those mutations are suggested to abrogate the interaction between MepS and sNlpI as indicated by the ITC experiments. However, the authors also state that monomeric sNlpI does not interact with MepS. Therefore, the authors need to provide evidence that the mutations L38A/C and R82E affect the direct interaction of MepS and NlpI and not merely prevent dimerization of NlpI. The model suggests that MepS also interacts with Prc, that then mediates the important steps in the degradation of the substrate. It is therefore not clear, why the authors did not include Prc in these experiments.

Thanks to the reviewer, constructing the ternary complex model is now included in Methods under a new section entitled “Docking analysis”.

Per the reviewer’s suggestion, we have performed additional AUC experiments and the results show that all the point mutants of NlpI are a dimer in solution like the wild-type NlpI; these results are presented in Results in the sub-section “TPR1 of NlpI is involved in MepS binding”.

We have included in the paper additional ITC analysis of MepS titrated with the Prc-NlpI complex and of MepS titrated with Prc alone in Fig. 6. The analysis shows no binding of sMepS to Prc alone. Furthermore, the calculated K_d values for sMepS binding to sNlpI alone and the sNlpI-Prc complex were 0.75 and 2.09 μM, respectively. These comparable K_d values suggest that the molecular surface of Prc in the NlpI-Prc complex may contribute little to the binding energy of MepS interaction. Therefore, for this work we focused on characterizing the interaction of MepS with NlpI but not Prc.

6) line 203 ff: The authors provide evidence that the Prc mutant L340G has an impaired activity and that the double mutant Prc-L340G/L245A almost lost activity. It would therefore be of interest to investigate the effect of the Prc-L245A mutant alone as well, since the difference in the activities between Fig. 6e and f seems small. Furthermore, can the authors comment on the fact that the two conserved residues L340 and L245 were mutated to G and A, respectively? Was the L340A mutant still active? It would make sense to mutate both leucines to the same amino acid.

We have now included SDS-PAGE analysis of Prc-L245A and L340A, in Fig. 9c and described the results in the rewritten last sub-section of Results; both mutants showed an impaired MepS degradation activity but not as dramatic as L340G. We have mutated L340 to G because of its location on a loop, presumably permitting a glycine mutation on the spot (by contrast, L245 is located on a short beta-strand, making a glycine mutation not possible). Moreover, based on the crystal structure we thought that the presence of any side chain, even a methyl group, at residue 340, might still mediate PDZ ligand sensing.

7) It makes sense to assume that the PDZ domain can adopt a closed conformation similar to the PDZ resting state in CtpB. However, the model that was constructed manually in Fig S5b is not useful in supporting this finding. It is no proof of this assumption and one can hardly read anything from this Figure. For reasons of comparison, it would also be interesting to know the dimensions of the movements (in angstrom) of the PDZ domains in CtpB and Prc (D1P).

We appreciate the reviewer's opinion about the usefulness of the resting model in Fig. S5b and have removed it from the manuscript.

And thanks to the reviewer's suggestion, the distance of the movements have now been added (now in Fig. S4b).

8) Is there evidence in the literature to support the hypothesis that the large movement of the PDZ domain, along with the unique structure of Prc may provide a mechanical pulling force for the degradation of substrate (lines 233-237)?

Large tethering movement causing structural deformation of the substrate protein has been proposed based on the structure of a serpin-protease complex by Hutington *et al.*, *Nature* 407, 923-926 (2000), which has now been cited in the revised manuscript and is discussed in the next response below.

9) The authors come up with a hypothesis for the “leverage” mechanism for substrate-degradation (line 254 ff.). Based on the evidence provided, however, it is hard to understand the rational basis of these assumptions.

Specifically, what is the structural basis for “substrate-deformation” (line 263) or that the substrate peptide chain is “pulled by rotational movement of the PDZ domain” (line 264) that allows cleavage by the catalytic dyad. These ideas haven’t been adequately developed anywhere in the manuscript, nor is there any evidence from literature provided that supports this mechanism. In my opinion, these assumption would require co-crystal structures of the complex with substrates in different states and/or extensive molecular dynamics simulations. Hence, I think it is too overstated to include in the manuscript.

We appreciate the reviewer’s criticism and acknowledge that the rational and structural bases for the proposed mechanism were not fully developed in the Discussion, as it should. In response to the reviewer’s comments, we have removed the mechanism statement from the Abstract; but we have attempted a rewrite in Discussion in order to cover the lever-like features of the sNlpI-Prc structure in this manuscript without overstatement:

“The overall structure of the sNlpI-Prc complex shows interesting features reminiscent of some specialized lever devices such as a wall-mount bottle opener and a yarn organizer. For example, the complex structure contains a fulcrum-like substrate-binding passage open on two sides: the side open from the exterior forms the substrate-docking cradle (the load); the other side leading to the interior center of the bowl-shaped Prc is attached to a hinged PDZ domain, which may exert effort/force by rotational movement (Fig. 9f). The substrate MepS is thought to dock into the cradle with its flexible C-terminal tail bound to the substrate-binding passage and the C-terminal end is captured by the resting PDZ domain from the opposite side. Binding of the substrate C-terminus may induce large rotational movement of the PDZ domain in Prc, which is likely stabilized by interaction with the NHD and the hinge residues. The movement of the substrate C-terminus-bound PDZ domain would drive the

translocation of the MepS polypeptide chain through the substrate-binding passage where the proteolytic site resides. By this process, to maintain covalent backbone linkage the globular structure of MepS trapped in the docking cradle may be partially unfolded, or structurally “deformed” before the threaded C-terminal substrate polypeptide chain is cleaved at the proteolytic site. After the release of the cleaved peptide fragment, the ligand-free PDZ domain resumes its resting position to capture the processed C-terminus of the substrate for the next round of the translocation/cleavage cycle. Future structural and computational studies are required to validate the proposed substrate degradation mechanism.

An earlier structural work on the serpin-protease complex has provided a specific example of structural deformation of bound substrate protein caused by large conformational change in the executor protein (Hutington *et al.*, 2000). In this case, the large conformational change of serpin is triggered by cleavage of its reactive loop by the substrate protease, which drives a large en bloc translocation by a distance of 70 Å of the covalently linked substrate to the opposite side of serpin (the executor); the resulting overlapping of the two structures causes structural deformation of the substrate. It is interesting to note that based on structural comparison the ligand-triggered rotational movement of the PDZ domain in Prc may occur by a comparable distance of ~50 Å (Supplementary Fig. 4b).”

We hope the above revised text is acceptable.

Minor points:

line 26: Abbreviation of out membrane (OM) is not defined.

This has been corrected in the new Abstract.

Line 26: Abbreviation of TRP is not defined.

It is now defined in the new Abstract.

line 65-66: The authors state that MepS and NlpI were expressed as soluble forms without their signal peptides. The methods state that NlpI is truncated before S20. It is not clear if the lipoprotein’s invariant cysteine is still part of the construct, which could lead to posttranslational modification and membrane anchoring.

Normally, this should not happen as the NlpI signal peptide region before S20 has been removed, hence including the lipobox Cys19, and the recombinant protein should be expressed like a typical cytoplasmic protein without posttranslational modification. The C-terminal His-tagged construct of MepS primary used in the paper is actually truncated before S28, which does not contain the Cys residue either; we have updated the construct info accordingly in Methods.

line 73-75: The authors state that reduced lysozyme is expected to have an unstructured C-terminus. Is there evidence of this in the literature or do the authors assume this?

We assumed this based on the crystal structure of lysozyme (PDB code 1DPX), which contains a disulfide bond formed between its N- and C-termini, by Cys6-Cys127.

line 114-115: It is not mentioned if the peptides that co-crystallized were added by the authors or co-purified.

Would it be possible to identify the peptide in the proteolytic groove and in the ligand binding site by mass spectrometry?

We have now specified in Results that the peptides were co-purified and co-crystallized serendipitously. We were unsuccessful in identifying the bound peptides by the method. The structures of CtpB also showed co-purified peptides modeled as poly-Ala, which were not identified in the paper.

line 141 ff: How does the activity of Prc in presence of sNlpI-QM compare to the activity of Prc alone? Quantification of the catalytic activity would be useful to support and to clarify these findings (see major point 2).

We have determined the specific activities of Prc without or with NlpI and NlpI-QM. The activity of Prc in the presence of NlpI-QM is weak and similar to that of Prc alone (0.98 and 0.58 nmole/min/mg, respectively); the result has been included in the revised paper to support that the QM mutant does not bind well to Prc.

line 160 ff: How was the model of the Prc-NlpI-MepS model constructed and validated? It is not mentioned in the methods section.

Our docked model was validated first by its consistency with previous finding that Prc recognizes the C-terminus of protein substrates, and further by site-directed mutagenesis as presented in the sub-section entitled “*TPRI of NlpI is involved in MepS binding*”. Per the reviewers’ request, docking analysis has now been described in Methods. Basically, MepS (PDB code 2K1G) was first docked to the crystal structure of sNlpI-Prc complex using the ClusPro 2.0 protein-protein docking server (<http://cluspro.bu.edu>) (Kozakov et al., 2010). Two of the top 10 docking models predicted by ClusPro placed MepS in the same prominent valley formed between the NlpI homodimer and Prc in the complex. Although the globular MepS in the valley were docked in different orientations, the revealed putative MepS-docking valley appeared convincing after manually performing rigid-body rotation, MepS was oriented such that its lipid-anchored N-terminal coil region is well accommodated in the complex structure while allowing its flexible C-terminus positioned precisely at the entrance of the substrate-binding passage of Prc (Figs. 6a,b).

line 193: “h7 of the NTD”... should this read NHD?

We thank the reviewer and have corrected the typo.

line 202: “Lke the wild type”... should read “Like the wild type”

We have corrected this typo also.

line 214: “structure of substrate-degrading Prc” is overstated as a catalytically inactive mutant was crystallized.

By the term we meant to specify that this is the structure of a PDZ protease that degrade but not merely cleave off the C-terminus of protein substrates. We have deleted the misleading wording and it now reads simply “structure of Prc”.

line 245: Is there anything known about the specific functions of NlpI, apart from its ability to bind to Prc and to “acts as a hub to coordinate the assembly of multi-protein complexes” as presented here? The authors should elaborate on this.

So far, NlpI is only known for serving as a substrate adaptor to bind to Prc and to facilitate MepS-degradation by Prc. It is currently unknown whether NlpI also participate in the assembly of other protein complexes. We have now included this notion in Discussion.

-----CIC-----

Reviewers' comments:

Reviewer #1 (Remarks to the Author):

The article by Ming-Yuan Su et al has improved from previous round with data added as requested. However, I still have concerns about the suitability of the manuscript - regardless of the authors response, the title, abstract and introduction do not provide enough explanation as to why this research is important to the wider community.

Major issues:

1) The title is now highly specific and it does not open to wider audience the biological significance or impact of the work.

I would at least modify the title to include at least "...degradation of E. coli peptidoglycan hydrolase ..." NIP1, MepS and PRC do not say much to others than specialists. which is fine for more specialized journal.

In the Abstract the impact is still missing. What is the importance of this specific system? In introduction there is reference to importance to bacterial viability. While this refers to MepS, MepM and MepH collectively. What is the importance of this particular system and how does it relate to MepM/H function?

2) The docked ternary model is not well-defined. it can be presented but it must be stated its just a putative model.

There are no restraints applied on docking and essentially it was manually docked.

test should read something like (line 190, p9)

" ...complex permits the construction of one possible model for interaction with MepS while detailed interactions cannot be verified without further experimental evidence and other models for ternary complex formation are possible"

3) the leverage/force/pull model on the importance of the PDZ domain hinge movement is highly speculative and lacks evidence.

Lines 274-278 are speculative and should be stated that "it can be hypothesized that"

line 281 - reference is missing for the original publication for NIP1 structure.

Lines 295-300 should be largely cut as well as the "force" model from Fig 9f. The whole paragraph should be shortened including the description of serpin complex few sentences should be enough.

partial unfolding or "deformation" or movement of the domain on substrate binding should be measured to verify these speculations.

Other points:

line 97-98 language, "MepS, which presumably acts"

line 125: Fig 4c does not have the crevice lined with polar residues? This is maybe OK, but perhaps a side view and a slice/cut-through view would be more informational on demonstrating the shape of the molecule. now there are 3 figures with not much information (Fig 4c). Perhaps the "crevice" could be indicated more clearly. (Cut through view might do this? plus indication what it is)

Catalytic residues refer to Suppl Fig 1 - they are not indicated there? please mark the residues.

line 160 typos/error - check. (similarity of On...??) more careful editing would be needed.

line 212 NIPI-deltaN is not defined anywhere - must be defined.

Also the mutations done - how where these selected actually?

line 213-215 p 10. These dont actually appear to be at the docked interface but at one end of it? is it possible these just destabilize the protein?

FIGURES.

Fig1 gels are now missing labels for the bands. (expect 1d).

Fig2

PDZ domain should be indicated.

Fig 3. order of labels is odd. in Fig3B the stereo figure is now too large - please make them viewable. Also why all the labels in the figure? are they all needed? looks messy. removed the ones not mentioned in the text.

Fig 8. Again stereo figure too large.

Fig 9c labels missing.

Fig9f - the model should be preferably simplified. the force-pull model is highly speculative and there is no evidence for it.

Or it must be clearly stated in discussion (which must be cut significantly by at least 50% that this is just a hypothetical mdel which requires verification by experiment)

As such there would still be several revisions needed before the manuscript is acceptable. Most importantly it is not clear neough if the biological impact is significant enough for the wider audience but with corrections paper might be acceptable.

Reviewer #2 (Remarks to the Author):

The manuscript has greatly improved and the authors responded to all my concerns and questions. This challenging work is technically sound and certainly of interest to the general readership of Nature Communications.

I only have one minor remark the authors should briefly comment on:

The authors suggest a mechanism for the complete degradation of MepS by Prc. Is anything known about cleavage products of MepS? e.g. how big are the peptide fragments and/or can those peptides be identified by any means? Of course, such experiments would likely be beyond the scope of the present study.

Furthermore, is there a hint of a recurring sequence of amino acids in MepS that could be specifically recognized by both, the substrate and PDZ binding pocket in Prc, mediating proteolytic cleavage? Those pockets are obviously quite similar and it is hard to conceive that the binding pockets do not have a more or less specific recognition sequence.

Reviewers' comments:

Reviewer #1 (Remarks to the Author):

The article by Ming-Yuan Su et al has improved from previous round with data added as requested. However, I still have concerns about the suitability of the manuscript - regardless of the authors response, the title, abstract and introduction do not provide enough explanation as to why this research is important to the wider community.

Major issues:

1) The title is now highly specific and it does not open to wider audience the biological significance or impact of the work.

I would at least modify the title to include at least "...degradation of E. coli peptidoglycan hydrolase ..." Nlpi, MepS and Prc do not say much to others than specialists. which is fine for more specialized journal.

Once more, we appreciate the reviewer's comments and suggestions. We have now changed the title accordingly to cover more biology and make it more appreciable to wider audience: "Structural basis of adaptor-mediated protein degradation by Prc, a tail-specific PDZ protease".

In the Abstract the impact is still missing. What is the importance of this specific system? In introduction there is reference to importance to bacterial viability. While this refers to MepS, MepM and MepH collectively. What is the importance of this particular system and how does it relate to MepM/H function?

We are grateful to the reviewer for pointing out the lack of the specific information on biological significance of the system in the Abstract and Introduction. We have now re-written the Abstract, which now provides the biological significance of the system while conforming to the word limit of 150 words. We have also modified the Introduction accordingly (highlighted in blue). Currently, it is not clear how the other two hydrolases MepM and MepH are regulated or whether they are also regulated by Prc-Nlpi-like protease-adaptor systems.

2) The docked ternary model is not well-defined. it can be presented but it must be stated its just a putative model. There are no restraints applied on docking and

essentially it was manually docked.

test should read something like (line 190, p9)

"...complex permits the construction of one possible model for interaction with MepS while detailed interactions cannot be verified without further experimental evidence and other models for ternary complex formation are possible"

Thanks to the reviewer's suggestion, we have added the suggested statement in the sentence.

3) the leverage/force/pull model on the importance of the PDZ domain hinge movement is highly speculative and lacks evidence.

Lines 274-278 are speculative and should be stated that "it can be hypothesized that"

We have used the wording suggested by the reviewer in the sentence to emphasize that the model is currently speculative without evidence.

line 281 - reference is missing for the original publication for NlPI structure.

The reference is now provided.

Lines 295-313 should be largely cut as well as the "force" model from Fig 9f. The whole paragraph should be shortened including the description of serpin complex few sentences should be enough.

partial unfolding or "deformation" or movement of the domain on substrate binding should be measured to verify these speculations.

In response to the reviewer's comments, we have largely shortened the concerned paragraph in Discussion. The previously elaborated descriptions on the speculated force model and deformation mechanism have been removed and now replaced by a sentence stating: "Details on how the lever-like features of the sNlPI-Prc complex are involved in degradation of MepS require further mechanistic characterizations."

Other points:

line 97-98 language, "MepS, which presumably acts"

Corrected accordingly.

line 125: Fig 4c does not have the crevice lined with polar residues? This is maybe OK, but perhaps a side view and a slice/cut-through view would be more informational on demonstrating the shape of the molecule. now there are 3 figures with not much information (Fig 4c). Perhaps the "crevice" could be indicated more clearly. (Cut through view might do this? plus indication what it is)

Per the reviewer's comment, we have now removed the three figures of Fig.4c. Since the role of the crevice is not discussed at all in this paper, we have also removed the short sentence describing the crevice in the main text.

Catalytic residues refer to Suppl Fig 1 - they are not indicated there? please mark the residues.

We have indicated the catalytic residues in the figure and the legend and labeled them with open triangles.

line 160 typos/error - check. (similarity of On...??) more careful editing would be needed.

We have corrected the typos/error and edited the main text thoroughly.

line 212 NIPI-deltaN is not defined anywhere - must be defined.

NlpI-deltaN was defined in Methods previously (line 345). We have now defined it also in the main-text sentence.

Also the mutations done - how where these selected actually?

line 213-215 p 10. These dont actually appear to be at the docked interface but at one end of it? is it possible these just destabilize the protein?

Our docking model suggests that NlpI may bind to the N-terminal helix of MepS via the juxtaposed helices h1 and TPR1b formed by the two subunits of the NlpI homodimer (Fig. 6b). Therefore, we assessed the role of the helices h1 and TPR1b of the NlpI homodimer in mediating MepS interaction by mutational analysis by ITC. Residue L38 of helix h1 (L38A or L38C) and R82 of helix TPR1b (R82E) of NlpI were chosen because they are close to the N-terminal helix of the docked MepS. Thanks to the reviewer, we have now added the description in the main text.

FIGURES.

Fig1 gels are now missing labels for the bands. (expect 1d).

The bands in Fig 1 gels are now labeled.

Fig2 PDZ domain should be indicated.

The two PDZ domains are now labeled in the figure.

Fig 3. order of labels is odd. in Fig3B the stereo figure is now too large - please make them viewable. Also why all the labels in the figure? are they all needed? looks messy. removed the ones not mentioned in the text.

We have resized the figure images and adjusted the labels accordingly as suggested.

Fig 8. Again stereo figure too large.

We have resized the figure images.

Fig 9c labels missing.

We have added the missing labels.

Fig9f - the model should be preferably simplified. the force-pull model is highly speculative and there is no evidence for it.

Or it must be clearly stated in discussion (which must be cut significantly by at least 50% that this is just a hypothetical model which requires verification by experiment)

As mentioned above, we have largely cut the discussion. Fig.9f now only serves to illustrate the level-like features of the complex structure.

As such there would still be several revisions needed before the manuscript is acceptable. Most importantly it is not clear enough if the biological impact is significant enough for the wider audience but with corrections paper might be acceptable.

We hope that the revisions made as listed above based on all of the reviewer's comments/suggestions are acceptable.

Reviewer #2 (Remarks to the Author):

The manuscript has greatly improved and the authors responded to all my concerns and questions. This challenging work is technically sound and certainly of interest to the general readership of Nature Communications.

I only have one minor remark the authors should briefly comment on:

The authors suggest a mechanism for the complete degradation of MepS by Prc. Is anything known about cleavage products of MepS? e.g. how big are the peptide fragments and/or can those peptides be identified by any means? Of course, such experiments would likely be beyond the scope of the present study.

We appreciate the reviewer #2's opinion that the ms is now suitable for the journal. Currently, it is not known of the size or the identity of the cleavage product fragments of MepS. The peptide products could theoretically be identified by mass spectrometry, which will be carried out in the future.

Furthermore, is there a hint of a recurring sequence of amino acids in MepS that could be specifically recognized by both, the substrate and PDZ binding pocket in Prc, mediating proteolytic cleavage? Those pockets are obviously quite similar and it is hard

to conceive that the binding pockets do not have a more or less specific recognition sequence.

We thank the reviewer for the excellent point, which has prompted us to examine the sequence of MepS more closely! We found that serine is the most abundant residue in MepS (11%) and nineteen XS dipeptide motifs are present throughout the sequence, which is quite interesting considering the similarly small pocket in the PDZ domain for the ligand C-terminal residue and the P1-binding pocket. Since the PDZ domain of Prc may recognize the C-terminal peptide sequence LSRS, it may be hypothesized that a degenerate dipeptide motif XS, where X may be any residues with a polar/charge side chain longer than serine, in MepS sequence may serve as the site for recognition/cleavage by Prc. Future studies will be carried out to test the hypothesis.

-----CIC-----